# Deep learning tools and modeling to estimate the temporal expression of cell cycle proteins from 2D still images

**Thierry Pécot** [1]*, **Maria C. Cuitiño**[2], **Roger H. Johnson**[3], **Cynthia Timmers**[4¤], **Gustavo Leone**[5]

**1** Rennes 1 University, SFR Biosit (UMS 3480 - US 018), Rennes, France, **2** Department of Radiation Oncology, Arthur G. James Hospital/Ohio State Comprehensive Cancer Center, Columbus, Ohio, United States of America, **3** Cancer Center, Medical College of Wisconsin, Milwaukee, Wisconsin, United States of America, **4** Division of Hematology and Oncology, College of Medicine, Medical University of South Carolina, Charleston, South Carolina, United States of America, **5** Department of Biochemistry, Cancer Center, Medical College of Wisconsin, Milwaukee, Wisconsin, United States of America

¤ Current address: Incyte Corporation, Wilmington, Delaware, United States of America
* thierry.pecot@univ-rennes1.fr

**Data Availability Statement:** Data availability All the images used in this study are available at https://data.mendeley.com/datasets/5r6kf37zd4/1.

## Abstract

Automatic characterization of fluorescent labeling in intact mammalian tissues remains a challenge due to the lack of quantifying techniques capable of segregating densely packed nuclei and intricate tissue patterns. Here, we describe a powerful deep learning-based approach that couples remarkably precise nuclear segmentation with quantitation of fluorescent labeling intensity within segmented nuclei, and then apply it to the analysis of cell cycle dependent protein concentration in mouse tissues using 2D fluorescent still images. First, several existing deep learning-based methods were evaluated to accurately segment nuclei using different imaging modalities with a small training dataset. Next, we developed a deep learning-based approach to identify and measure fluorescent labels within segmented nuclei, and created an ImageJ plugin to allow for efficient manual correction of nuclear segmentation and label identification. Lastly, using fluorescence intensity as a readout for protein concentration, a three-step global estimation method was applied to the characterization of the cell cycle dependent expression of E2F proteins in the developing mouse intestine.

## Author summary

Estimating the evolution of protein concentration over the cell cycle is an important step towards a better understanding of this key biological process. Unfortunately, experimental designs to monitor proteins in individual living cells are expensive and difficult to set up. We propose instead to consider 2D images from tissue biopsies as snapshots of cell populations to reconstruct the actual protein concentration evolution over the cell cycle. This requires to accurately localize cell nuclei and identify nuclear fluorescent proteins. We take advantage of the powerful deep learning technology, a machine learning approach

The training datasets for nuclei segmentation are available at https://github.com/tpecot/NucleiSegmentationAndMarkerIdentification/tree/master/UNet/datasets/nucleiSegmentation_E2Fs for the U-Net architecture and at https://github.com/tpecot/NucleiSegmentationAndMarkerIDentification/tree/master/MaskRCNN/datasets/nucleiSegmentation_E2Fs for the Mask R-CNN architecture. The training datasets for nuclei segmentation and marker identification are available at https://github.com/tpecot/NucleiSegmentationAndMarkerIDentification/tree/master/InceptionV3/trainingData for the Inception-V3 architecture. The images used for the evaluation and the ground truth are available at the same locations as the training datasets. The intensity 2D histograms used to estimate the E2Fs accumulation over the cell cycle are available at https://github.com/tpecot/EstimationOfProteinConcentrationOverTime/tree/master/data. Software availability The codes used to train and process deep learning approaches for nuclei segmentations and marker identification are available at https://github.com/tpecot/NucleiSegmentationAndMarkerIDentification. Archived code at time of publication: https://doi.org/10.5281/zenodo.4619243 [50] License: GPL3 The Octave code used to estimate the E2Fs accumulation over the cell cycle is available at https://github.com/tpecot/EstimationOfProteinConcentrationOverTime. Archived code at time of publication: https://doi.org/10.5281/zenodo.4639800 [51] License: GPL3 The Java code of the Annotater plugin and the plugin are available at https://github.com/tpecot/Annotater. Video tutorials to show how to use the Annotater are available at the same location. Archived code at time of publication: https://doi.org/10.5281/zenodo.4639802 [52] License: GPL3.

**Funding:** This work was funded by a Chan Zuckerberg Initiative (https://chanzuckerberg.com/) DAF grant to T.P. (2019-198009), an NCI grant (R50 CA211529) to C.T., Advancing a Healthier Wisconsin Endowment to G.L. and a Dr. Glenn R. and Nancy A. Linnerson Endowed Fund to G.L.. The funders had no role in study design, data collection and analysis, decision to publish, or preparation of the manuscript.

**Competing interests:** The authors have declared that no competing interests exist.

which has revolutionized computer vision, to achieve these challenging tasks. Additionally, we have created an ImageJ plugin to quickly and efficiently annotate images or correct annotations, required to build training datasets to feed the deep convolutional neural networks.

This is a *PLOS Computational Biology* Methods paper.

## Introduction

Automatic image analysis is at the core of human and animal tissue-based research. However, quantitation of morphological features or fluorescent labeling in intact mammalian tissues still remains a challenge. The densely packed nuclear aggregates that characterize many of these tissues, the extensive variability across different tissue types, and the continuously increasing number of imaging modalities are some of the many variables that make tissue biological quantification an extremely difficult task. Over the last decade [1–4], deep learning has brought artificial intelligence to the forefront of image-based decision making. In particular, deep convolutional neural networks have demonstrated their superiority for image segmentation [1, 2]. These approaches have also outperformed the traditional approaches used in microscopy, such as watershed for nuclei or cell segmentation [5–9]. However, this machine learning-based approach requires large amounts of annotated data and new strategies have to be developed to process highly complex biological objects acquired with different modalities by considering small training datasets. In this paper, we propose a series of deep learning-based approaches to precisely segment nuclei and to identify fluorescently labelled cells in order to analyze the evolution of cell cycle dependent E2F protein concentration in mouse tissues.

E2Fs are major regulators of the cell cycle. The members of this family of transcription factors are categorized into three subclasses in mammals: canonical activators (E2F1–3A/B), canonical repressors (E2F4–6), and atypical repressors (E2F7–8) [10–13]. Adding to the body of literature on E2F-dependent transcriptional activity in vivo, our lab previously provided quantitative evidence on the temporal expression of representative activator (E2F3A), canonical repressor (E2F4) and atypical repressor (E2F8) family members during embryonic development [14], all of which have been shown to be of major importance [15–18]. To establish the temporal expression profiles of the three sentinel E2Fs, we used an E2F3A specific antibody and generated MYC-tagged E2F4/E2F8 knock-in mice. In addition to fluorescence labeling of E2F3A, E2F4 and E2F8, 5-ethynyl-2'-deoxyuridine (EdU) and Histone H3 S10 phosphorylation (pH3) were used to identify S, G2 and M phases. Images of eight different combinations of markers were acquired from sections of the developing mouse intestine using confocal and widefield microscopy (see S1 Fig). The data analysis pipeline consisted of i) nuclear segmentation with a deep learning approach [19], ii) nuclear marker identification by thresholding, and iii) estimation of E2F concentrations over the cell cycle from 2D intensity histograms.

In this manuscript, we propose and evaluate alternative methods to quantify nuclear protein levels which result in an improved automated pipeline with greatly reduced requirement for interactive manual corrections. Using a small training dataset composed of 2D still images (with and without various forms of data augmentation), we first evaluate five different deep

learning strategies to segment nuclei in microscopic images of embryonic mouse intestinal epithelium. We also design post-processing methods to improve nuclear segmentation. We then propose another deep learning-based approach for identifying nuclear markers in the epithelial cells and demonstrate the superiority of this method to the usual threshold based method [20, 21]. Additionally, we create an ImageJ plugin [22, 23] named *Annotater* to specifically and efficiently correct nuclear segmentation and marker identification, ensuring that nuclear features are accurately quantified. Next, these image features extracted from 2D still images are used to perform a temporal analysis of E2F protein concentration over the cell cycle. Based on three mathematical assumptions grounded in cell biology, we initialize the temporal evolution of E2F concentrations using a graph optimization method. Cell cycle markers are then used to temporally register E2F proteins' concentration with respect to cell cycle phase. The global estimation of the protein concentration of E2F3A, E2F4 and E2F8 through the cell cycle is defined as an assignment problem and solved with the Hungarian algorithm [24, 25]. This approach is extensively evaluated with simulated data. Finally, we directly estimate the temporal evolution of E2F concentrations without using marker identification. In addition, we evaluate the impact of using different amount of images in the training datasets for nuclei segmentation on the estimation of E2F concentrations over the cell cycle.

## Results

### Mask R-CNN is the optimal deep learning approach for segmenting nuclei in cross-sectional images of complex tissues

Over the last decade, deep learning has revolutionized computer vision [1–4]. Over the years, several deep learning approaches have been successfully applied to segment cell nuclei [19, 26, 27]. More recently, the 2018 data science bowl [28] attempted to definitely solve this problem in 2D by posing the challenge: Create an algorithm for automated nuclei detection. Although impressive results were obtained for different light microscopy modalities and a variety of nuclear stains [7], nuclear segmentation from complex tissues such as intestinal epithelium still represents an unusually challenging problem. As is the case for most biological imaging studies due to natural variability in the objects of interest that are captured using dissimilar imaging modalities, no annotated data is available for this specific application. Consequently, we set out with the goal of designing a robust approach that would lend itself to routine use in the typical biology laboratory. We evaluated five different deep learning approaches for nuclear segmentation: U-Net [27, 29], Inception-V3 [30], Mask R-CNN [31], Stardist [8] and CellPose [9]. U-Net, perhaps the most-used deep convolutional neural network for biomedical data, is composed of an encoder part, used to capture image features, and a decoder part to estimate a class at each pixel of the input images. Inception-V3, which was designed to identify objects in images, is a deep convolutional neural network that only estimates one class given an input image. When using this architecture for nuclear segmentation, inputs are defined as image patches and the output corresponds to the class at the patch center. Inception-V3 is much slower than U-Net for both training and processing (S1 Table), as a decision is only made for the central pixel of the input image patch, in contrast to a decision being made at each pixel of the input image as it is in U-Net. For both U-Net and Inception-V3, three classes are defined: inner nuclei, nuclei contours, and background [19, 32]. Individual nuclei are then obtained by subtracting the nuclei contours from the inner nuclei (see Materials and methods). To improve performance, we developed a post-processing method that we call corrected watershed, wherein the results obtained with the U-Net or Inception-V3 network are combined with those produced by the watershed method [5] (see Materials and methods). Mask R-CNN, Stardist and Cellpose are instance segmentation approaches, *i.e.* they directly estimate

individual objects. These three methods first extract image features using a backbone convolutional neural network. Stardist identifies individual objects by predicting the distances to object boundaries with a fixed set of rays, ending up with a set of polygons for a given input image, corresponding to nuclei. Cellpose predicts an alternative representation of the nuclei masks, the equilibrium distribution of a heat-diffusion simulation with a heat source placed at the center of the mask. Mask R-CNN relies on a Region Proposal Network (RPN) to submit subregions of the input image. A fully connected neural network then defines a class and a bounding box for the input subregions, and a convolutional neural network generates a segmentation mask for the same input subregions. With these three approaches, the output corresponds directly to individual nuclei. The five approaches and their use are described in more details in the Materials and methods. Pre-processing, normalization, optimization, data augmentation, transfer learning and post-processing are summarized for each method in S2 Table. The number of images and nuclei used in the training and validation datasets are shown in S3 Table. Two different modalities, confocal and widefield images are used in this study, which allows us to test the genericity of the deep learning approaches for nuclei segmentation.

As proposed by Caicedo *et al*. [32], we use the F1 score with respect to the Intersection over Union (IoU) to compare performance of the three deep learning approaches (see Materials and methods). There are two aspects to performance: 1) the ability to correctly identify all nuclei in an image, which is the same as detecting the correct number of nuclei, and 2) the accuracy of the nuclear contours created in the output. This second aspect of performance is quantified by the F1 score over a range of IoU values. The IoU of two objects is the ratio of the intersection to the union of their areas. With an IoU of at least 0.5 (within the range 0.5–1.0), only two nuclei, one from the ground truth as determined by a pathologist and one estimated with a given method, can be paired. The F1 score for an IoU equal to 0.5 can be used to assess the ability of a method to accurately identify all nuclei. F1 scores for IoU thresholds in the range 0.5–1.0 reflect the localization precision of the segmentation, which means the accuracy of the defined nuclear contours.

The three instance segmentation approaches, *i.e.* Stardist, Cellpose and Mask R-CNN, show similar performance (see S4–S6 Figs and S5 and S6 Tables), with a lower accuracy for Cellpose. This approach has been designed to segment cells with cytoplasmic markers and is therefore less efficient with nuclear markers. Stardist shows a slightly better F1 score for an IoU threshold equal to 0.5 for confocal images than Mask R-CNN (0.864 vs 0.858), suggesting that a few more nuclei are identified with Stardist. However, the F1 scores for higher thresholds of IoU obtained with Mask R-CNN are clearly higher than those obtained with Stardist ($IoU$ = 0.75, 0.458 vs 0.434 for confocal images and 0.460 vs 0.389 for widefield images). This demonstrates that Mask R-CNN better localizes the nuclei areas than Stardist. Of note, while pooling together confocal and widefield images leads to a better accuracy for all three methods, data augmentation does not improve performance for Stardist and Cellpose while it clearly ameliorates Mask R-CNN precision. This could suggest that Stardist and Cellpose are more sensitive to noise when the training dataset is small and the nuclei are densely packed.

In the following, only Mask R-CNN is compared to U-Net and Inception-V3. Fig 1 shows the superiority of the Mask R-CNN method for both modalities. The results represent the best performance of each deep learning method from among the algorithmic variants that invoke combinations of transfer learning, data augmentation and watershed correction as discussed below in connection with S2–S4 Figs. S5 and S6 Tables show the F1 score obtained with IoU = 0.5 and IoU = 0.75 when considering all algorithmic variants for each method.

As shown in Fig 1a and 1b, and emphasized in the magnified images of Fig 1c and 1d, the U-Net method produced the worst results. Data augmentation, a process that artificially and

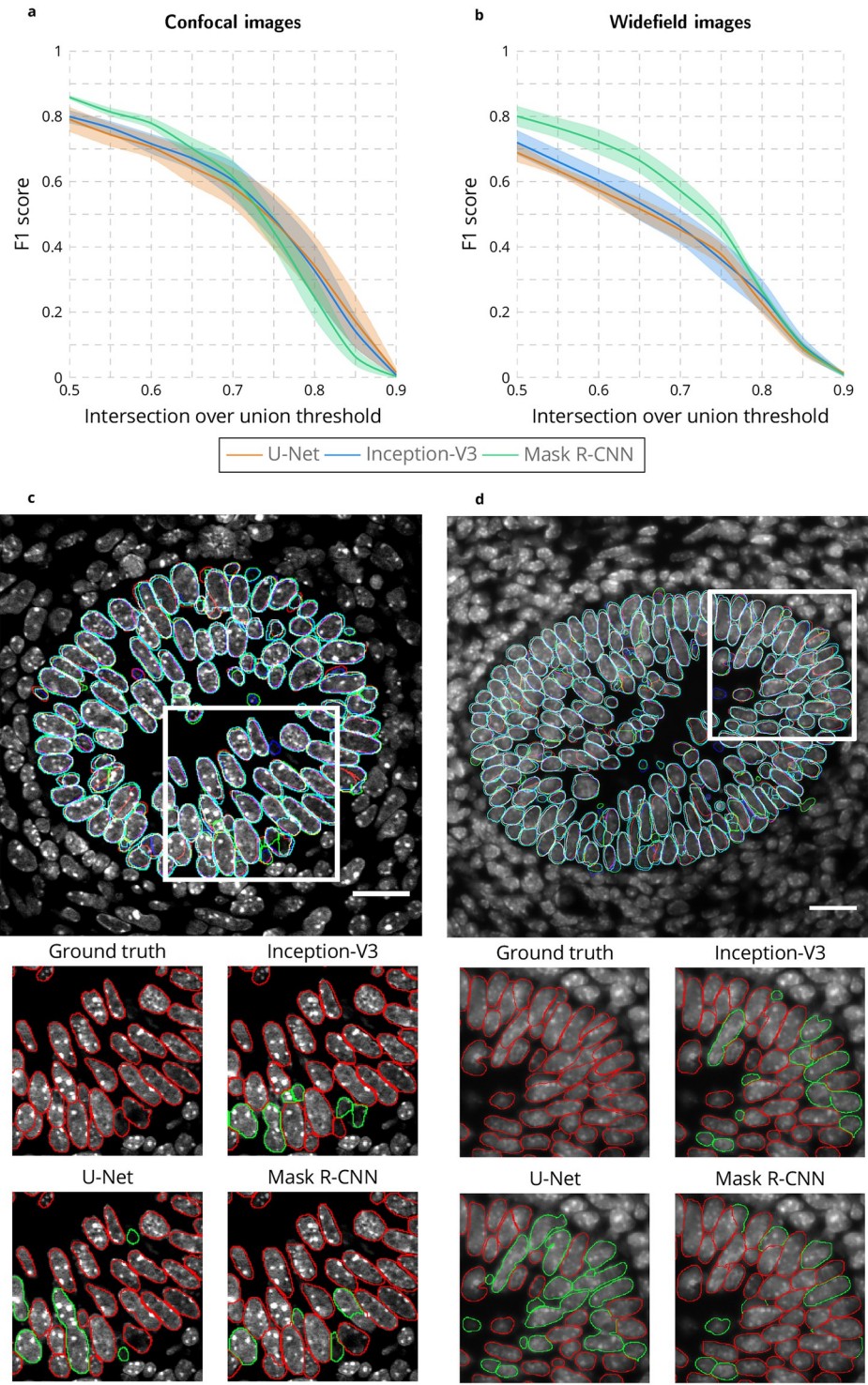

**Fig 1. Comparison of U-Net, Inception-V3 and Mask R-CNN for nuclear segmentation. a-b** F1 score for range of IoU thresholds obtained with the U-Net, Inception-V3 and Mask R-CNN approaches for confocal **a** and widefield **b** images. Lines correspond to average F1 score over the three tested images while the shaded areas represent the standard error. **c-d** Segmented nuclei obtained with the U-Net (blue), Inception-V3 (green) and Mask R-CNN (cyan) as well as the ground truth (red) for confocal **c** and widefield **d** images. White inset rectangles in top images are shown at higher magnification below for the ground truth and each approach. In high-mag images, correctly identified nuclei are outlined in red, incorrectly in green. Scale bar = 20μm.

massively increases the size of the training dataset by applying mathematical operations such as adding noise and rotating or flipping the images (see Materials and methods), clearly improves the segmentation accuracy (S2c and S2d Fig). However, pooling together confocal and widefield images in the training dataset produces worse segmentation accuracy compared to training on only confocal or only on widefield images. This demonstrates the inability of the U-Net approach to generalize nuclear segmentation with a small training dataset.

In contrast, the Inception-V3 approach obtains better results when both modalities are used in the training dataset (S3 Fig). While for Inception-V3 the training dataset was not increased when applying data augmentation because the computation time is already long, the high degree of overlap between input patches to which mathematical operations are applied (see Materials and methods) produces an effect similar to data augmentation, explaining in part the improved performance of the Inception-V3 approach relative to U-Net. Because Inception-V3 estimates the class one pixel at a time, both training and processing are highly compute intensive and not feasible in practice (S3d Fig). The proposed corrected watershed post-processing improves the results obtained for both U-Net and Inception-V3 (S2e, S2f, S3e and S3f Fig and S5 and S6 Tables).

The Mask R-CNN method obtains the best nuclear segmentation for both modalities (Fig 1), in large part due to the improvement realized through data augmentation and in a lesser extent to the transfer learning from the coco dataset [33], (Fig 1a and 1b). Combining confocal and widefield images in the training dataset improves the results, especially for confocal images (S4c–S4e Fig). This demonstrates how Mask R-CNN can benefit from an increase in training dataset size (from 4847/1619 nuclei in the training/validation dataset for confocal images alone to 15541/4051 nuclei in the training/validation dataset when confocal and wide-field images are pooled together) even if the data are not uniform, coming from different sources. Of note, combined with massive data augmentation, transfer learning from the coco dataset really improves accuracy when considering confocal images only (S4e Fig compared to S4c Fig) or widefield images only (S4f Fig compared to S4d Fig), while it does not drastically improve the accuracy when confocal and widefield images are pooled together (S4c–S4f Fig). This suggests than considering images from different modalities has a similar effect for convergence to a plateau with Mask R-CNN than transfer learning with a pre-trained model. Moreover, training and processing are fast (S4d Fig), and the results do not require any post-processing. One limitation of the Mask R-CNN method compared with U-Net and Inception-V3 is an inferior boundary localization accuracy for confocal images, as demonstrated by the lower F1 scores for IoU threshold values greater than 0.75 (Fig 1a). Due to slightly larger nuclear masks (cyan boundaries in Fig 1c), this limitation is more than compensated for by Mask R-CNN's higher true positive nuclei identification rate in most biological applications. While the performance of Mask R-CNN is impressive, its accuracy is not perfect and may be insufficient for many applications. Therefore, we designed the ImageJ plugin Annotater (see Materials and methods), a tool that allows users to efficiently correct the nuclear segmentation.

## Deep learning improves identification of fluorescent nuclear markers

After completion of nuclei segmentation with the DAPI channel, it is possible to use the fluorescence in the other channels to extract information of interest about the cells. E2Fs positive/negative status can be evaluated that way, as well as EdU and pH3 patterns. EdU and pH3 are cell cycle markers that show evolving patterns along the cell cycle [14]. EdU is diffuse during first half of S phase and becomes punctate during second half of S phase. pH3 is first punctate during second half of S phase and G2, and becomes diffuse during mitosis. Typically, a

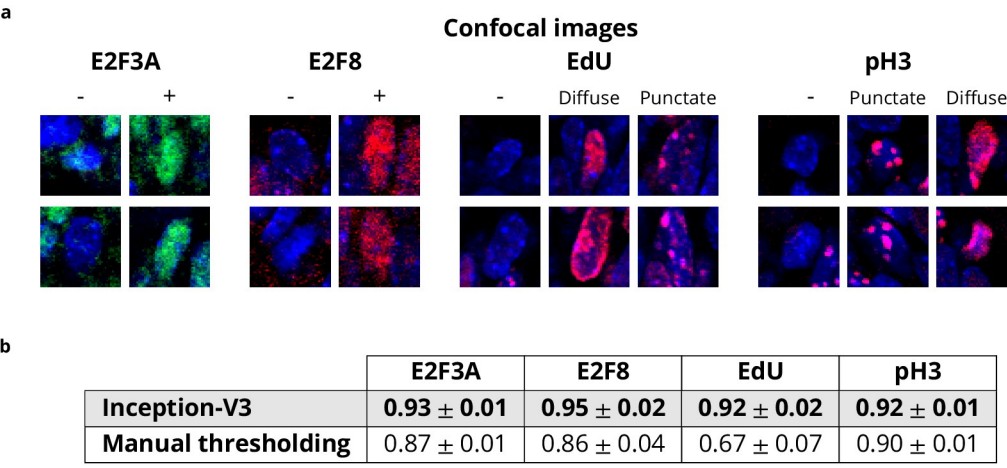

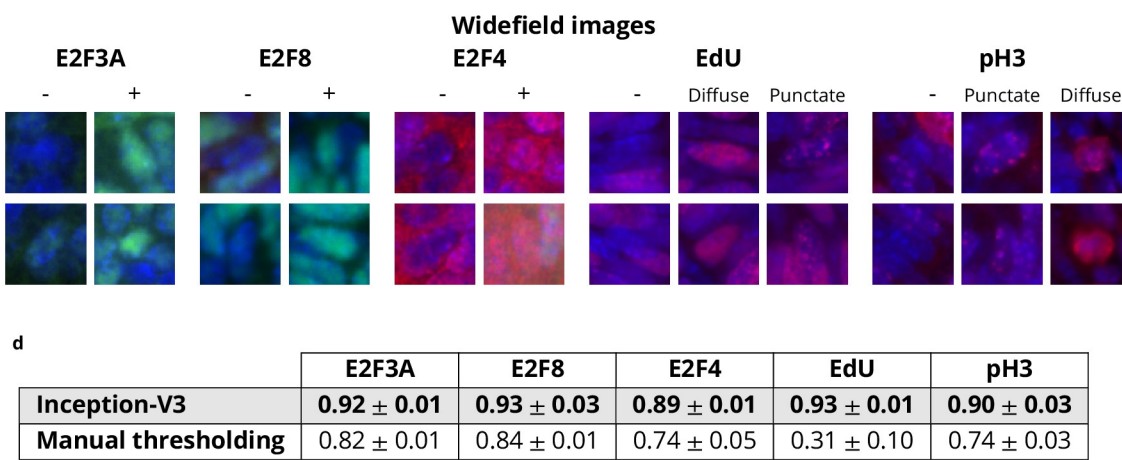

**Fig 2. Comparison between Inception-V3 and manual thresholding for marker identification. a-b** Top rows (images): examples of E2F3A-, E2F8- and E2F4-positive and -negative nuclei, as well as EdU- and pH3-negative, diffuse and punctate nuclei in **a** confocal and **b** widefield images. Bottom rows (bar graphs): Accuracy obtained with the Inception-V3 and manual thresholding approaches for marker identification of E2F3A, E2F8, E2F4, EdU and pH3 in **a** confocal and **b** widefield images.

thresholding procedure is applied to identify nuclear markers [20, 21], but this approach is not always accurate, especially when different patterns of fluorescence over a wide range of intensities are involved as is the case for EdU and pH3 (see diffuse and punctate patterns in Fig 2). To improve accuracy, we tested a deep learning approach for nuclear marker identification. As the goal is not to identify regions but to make a decision for each nucleus regarding the presence/absence of an E2F or the diffuse/punctate/absence of EdU or pH3, the instance segmentation and U-Net approaches are not suitable. In contrast, with an input defined as an image patch centered on each nucleus, the Inception-V3 architecture is appropriate to decide about the presence, potentially in diffuse or punctate state, or absence of a marker. In addition to data augmentation, we also define a so-called pixel-based training dataset (as opposed to nuclei-based training dataset) that includes the input patches centered at each pixel belonging to the nuclei (see Materials and methods). This strategy has a similar effect to data augmentation, as it drastically increases the training dataset. Although DAPI staining is different from

the nuclear markers used to identify the E2Fs, EdU and pH3, the images are acquired simultaneously, so the image features captured by the Inception-V3 method for nuclear segmentation are potentially meaningful to identify nuclear markers. Consequently, we also perform transfer learning from the nuclear segmentation (see Materials and methods). To easily set the threshold for marker identification, we designed an interface in the Annotater that we used to obtain the results shown in Fig 2 for manual thresholding.

As shown in Fig 2, compared to manual thresholding, the Inception-V3 approach provides better performance for each marker in both modalities. Manual thresholding achieves a relatively good performance for E2F3A and E2F8 identification in both modalities, and for pH3 in confocal images. The latter might appear surprising, but the two different patterns for pH3 in confocal images are different enough to allow a strategy based on the thresholded area in the nuclei (see Materials and methods) to lead to satisfying results (Fig 2a). However, the results for E2F4, Fig 2b, are not as good: Because E2F4 is also cytoplasmic, the extra-nuclear fluorescence confounds the thresholding decision. Finally, thresholding clearly fails to identify the two different patterns (diffuse, punctate) of EdU in images from both modalities as well as the patterns of pH3 in widefield images. In contrast, the Inception-V3 approach yields accuracies greater than 90% for all markers except E2F4 (89%) (Fig 2). As shown in S7 Table, the use of the pixel-based training datasets does not significantly improve marker identification for confocal images, but does improve performance on E2F3A, E2F4 and pH3 markers in widefield images (see S8 Table). Additionally, the transfer learning from the nuclear segmentation slightly improves the results for all markers in both modalities. Computation (after training) is fast because only one decision is made per nucleus (S9 Table). Overall, this study demonstrates the remarkable accuracy of the deep learning-based approach in identifying cells that are positive for the tested nuclear markers, not only when the marker is both nuclear and cytoplasmic, but also when it exhibits different labeling patterns, such as diffuse and punctate. Given such a high level of accuracy, correcting the results with the Annotater plugin takes only a short time.

## Estimation of E2F accumulation over the cell cycle from 2D still images

Estimating cell cycle progression is an important biological question and can be answered by using images [34, 35]. Unfortunately, these machine learning approaches require annotated data to be processed. Without training data, the most straightforward way to assess the evolution of a protein's concentration over the cell cycle would be to monitor the expression level in every cell through all phases of the cycle. However, this is not possible in the proliferative tissue of a living animal. Instead of observing one cell, we propose to observe a large population of cells, each corresponding to a snapshot characterizing the cell state at a particular time during the cell cycle, and to reconstruct protein concentration as a function of time in the cell cycle by combining all these snapshots. As fluorescence intensity is proportional to protein concentration [36, 37], we define quantized levels of intensity for E2Fs (see S7 Fig and Materials and methods). In each individual image, we measure the average fluorescence intensity for each positive cell. Within each image, the range of intensity from the lowest average intensity to the highest is divided into bins that define the levels of intensity. On the other hand, EdU and pH3 markers are defined by their diffuse and punctate states, which change as a function of cell cycle phase. As shown in our previous work [14], EdU shows a diffuse pattern during the first half of S phase and a punctate pattern during the second half of S phase. pH3 shows a punctate pattern during the second half of S phase and G2 and a diffuse pattern during mitosis. These two markers allow us to register E2Fs expression with respect to the cell cycle. We propose to globally estimate the E2F concentration's evolution over the cell cycle in three steps: i) initialization, ii) cell cycle registration and iii) global optimization. In this paper, as in our previous

manuscript [14], the phrase "protein accumulation" is synonymous with "protein concentration" and connotes this balance between production and degradation, regardless of whether the protein concentration is increasing or decreasing. Also in this paper, the fluctuation over time of a quantity such as fluorescence intensity or protein accumulation is referred to as "evolution," which is therefore synonymous with "time course."

We use the term "initialize" to mean creation of a first estimate of protein accumulation over the cell cycle: a graph of fluorescence intensity vs. time. To initialize all combinations of markers from the images (see S1 Fig), we make three fundamental assumptions:

1. The number of cells in a given phase of the cell cycle is proportional to the duration of that phase.

2. Temporal evolution of protein accumulation is similar in all observed cells (there are no subpopulations with different cell cycle evolution).

3. Concentrations of E2Fs evolution can be represented by concave downward parabolas, *i.e.* they increase from 0 to their maximum and then decrease from this maximum to 0.

These assumptions are validated by the following biological statements:

1. Proliferation in intestinal epithelium is asynchronous, so each cell cycle phase is observed at a frequency proportional to its duration.

2. All cells in intestinal epithelium are proliferating and undergo cell cycle at a uniform rate.

3. Cell cycle-regulated proteins first accumulate over time and are then degraded.

We use 2D histograms to initialize protein accumulation (S8 Fig and Materials and methods). In these histograms, the first axis corresponds to the intensity of an E2F, while the second axis corresponds to either another E2F intensity or the states of EdU or pH3 staining. From these histograms, we define graphs with costs associated with edges reflecting the possible intensity/pattern transitions of the E2Fs, EdU and pH3. The sequence of edges that goes through all vertices with minimum cost is used to initialize the evolution of protein accumulation over time (S9 Fig and Materials and methods). The initializations obtained for all eight different combinations of markers are shown in S10 Fig. As EdU and pH3 labeling patterns with respect to the cell cycle are known, we then register E2Fs with respect to EdU and pH3 through a circular permutation (see Materials and methods). We use the same approach to refine EdU labeling during the cell cycle (see Materials and methods). The protein accumulations after registration are shown in S11 Fig. We finally model the global estimation of E2F accumulation over the cell cycle as an assignment problem. We use the Hungarian algorithm [24, 25] to successively estimate the individual accumulation of each E2F over the cell cycle given the EdU and pH3 patterns as constraints (see Materials and methods). We also add a local constraint to only allow growing substitutions for E2Fs (see Materials and methods). The final result is shown in Fig 3, depicting the successive waves of E2F3A, E2F4 and E2F8 over the cell cycle.

We then simulate data to evaluate the validity of our approach (see Materials and methods). In summary, E2F3A and E2F8 are randomly generated to have a concentration that shows a concave downward parabola representation while E2F4 is randomly generated to have a concentration that shows two successive concave downward parabola representations, with random durations for each intensity level for all three E2Fs. EdU and pH3 are also randomly generated to show a succession of punctate/diffuse or diffuse/punctate patterns. Noise is randomly added to corrupt from 0% to 50% of temporal bins. For each set of parameters, 5 different simulations are randomly generated. The performance is evaluated by measuring the mean squared error (MSE) between the generated and estimated concentrations of E2Fs, EdU and

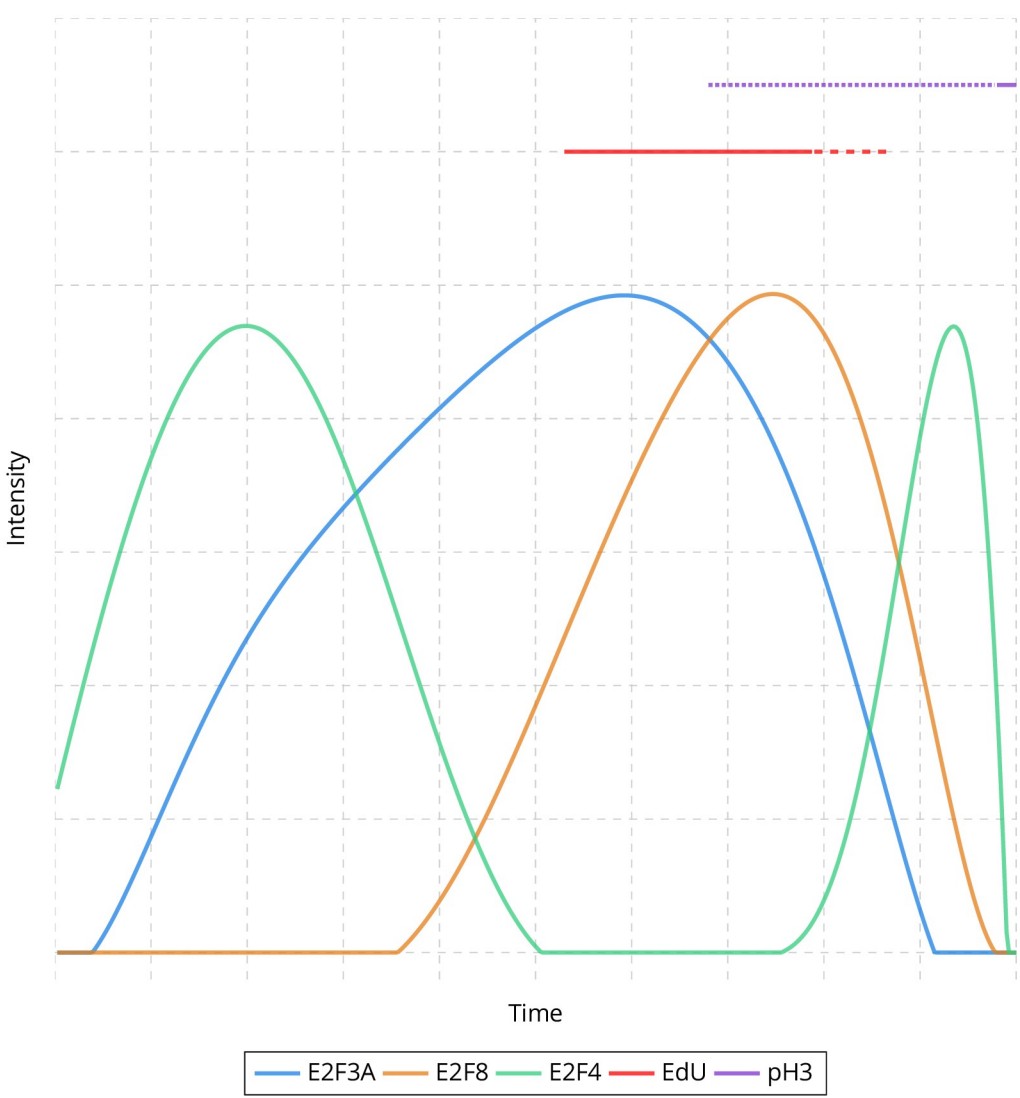

**Fig 3. Estimation of E2Fs evolution over the cell cycle.** Temporal evolution of E2F3A, E2F8 and E2F4 protein accumulation, as well as EdU and pH3 patterns over the cell cycle in mouse intestinal epithelium.

pH3. A first set of simulations estimates the influence of the number of samples, corresponding to the number of mice, and shows that the MSE drastically decreases from 1 sample to 2 samples (see S12 Fig), suggesting that 3 mice for E2F3A, E2F8, EdU and pH3, and 2 mice for E2F4 should lead to a satisfying estimation. A second set of simulations evaluates the influence of the number of bins used for time. As shown in S13 Fig, the MSE is low for a range of 20 to 100 bins and drastically increases for 125 bins, validating the choice of 100 bins for time. A last set of simulations estimates the impact of the number of bins used for intensity. The MSE is quite stable for a range of 2 to 5 bins but ramps up with 10 bins (see S14 Fig). This confirms our choice to use 4 bins, which is also widely used in both diagnostic pathology and biomedical research [38, 39]. Fig 4 summarizes the evaluation with simulations when considering the same parameters (number of samples, intensity and time bins) than those used for real data with up to 50% of the simulated data corrupted with noise.

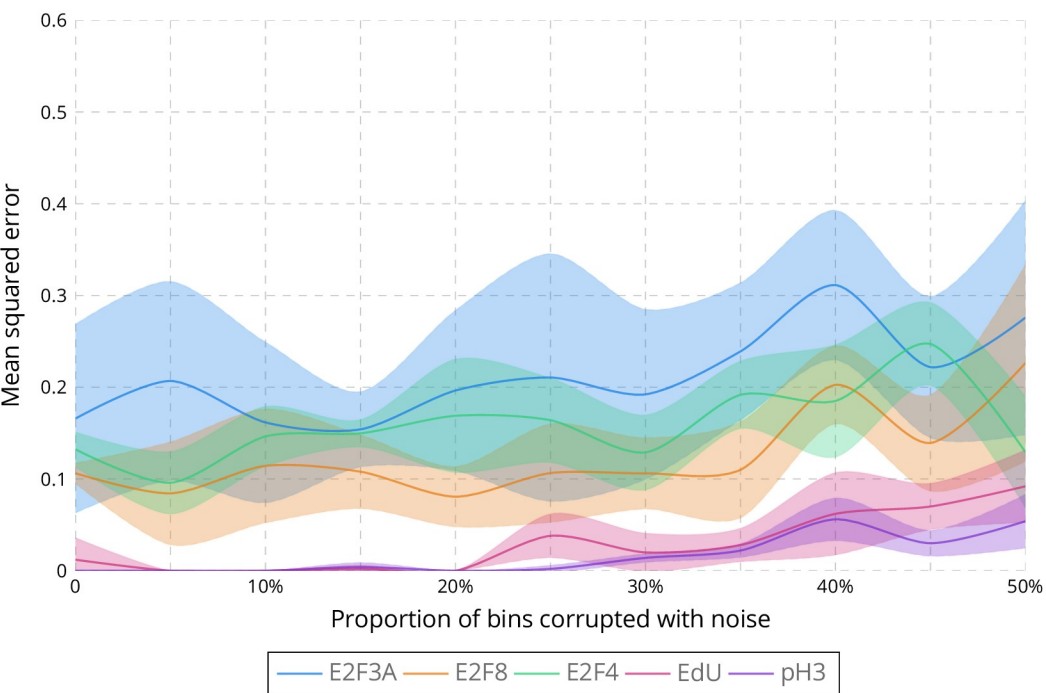

**Fig 4. Evaluation of the estimation of protein concentration over the cell cycle on simulated data.** Mean squared error between the estimated and simulated concentrations of E2F3A, E2F8, E2F4, EdU and pH3 when corrupting up to 50% of the simulated data with noise and considering 3 samples for E2F3A, E2F8, EdU and pH3, 2 samples for E2F4, 100 bins for time and 4 bins for intensity. The lines correspond to the average mean squared error while the areas represent the standard error.

## Estimation of E2F accumulation over the cell cycle with respect to nuclei segmentation

In the previous section, we used the marker identification to identify negative cells for E2Fs and the two patterns for EdU and pH3. We now want to assess if this step is required or not. Fig 5a shows the estimation of E2Fs concentration when considering only positive/negative states for EdU and pH3. While the estimated concentration slightly changes, especially the second peak of E2F4, the successive waves of E2Fs are preserved in time. This opens the possibility to a more direct concentration estimation where a nuclei segmentation is performed first and the intensity is then binned into four levels for E2Fs channels and two levels for EdU and pH3 (see Materials and methods). Fig 5b depicts the estimated concentrations for E2Fs with a manual segmentation for nuclei. The estimated concentrations are more degraded than with a better estimation of EdU and pH3 states, but E2Fs still show 4 successive peaks which are temporally well estimated, with the exception of the first E2F4 peak which is slightly delayed. This approach allows to evaluate the impact of nuclei segmentation on the concentration estimation over the cell cycle. We estimate the E2Fs concentration with the nuclei segmentation obtained for the confocal images with a Mask R-CNN model trained with the widefield images and a manual segmentation for the widefield images (see Fig 5c), and conversely (see Fig 5d). These estimations are actually close to the one obtained with manual segmentation, *i.e.* a degraded accuracy as shown in Fig 5e, but a well estimated temporality for the successive waves of E2Fs concentrations. This illustrates the robustness of our approach with respect to nuclei segmentation to temporally estimate the waves of E2Fs over the cell cycle.

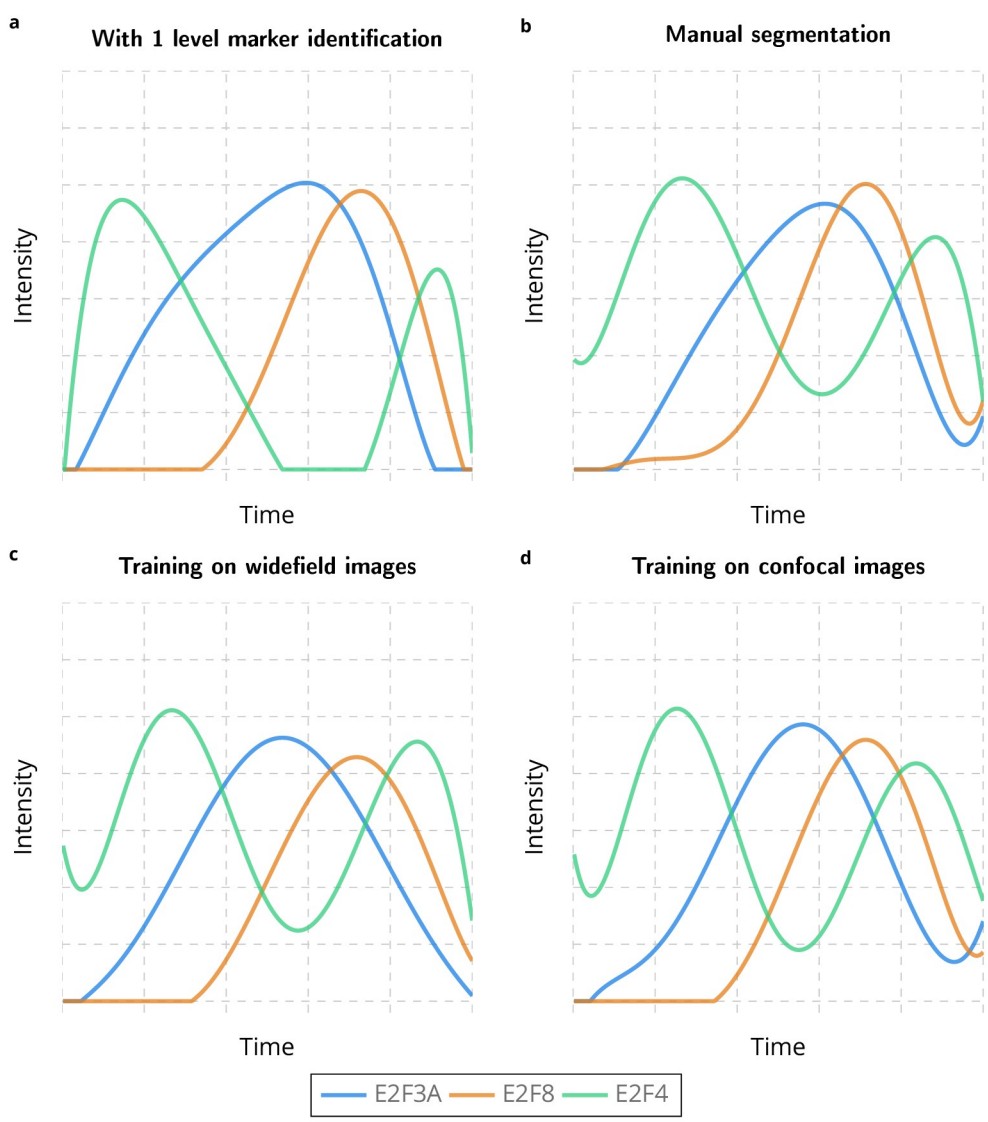

| | Manual segmentation | Training on widefield images | Training on confocal images |
|---|---|---|---|
| **MSE for E2F3A** | 0.914 | 0.881 | 0.985 |
| **MSE for E2F8** | 0.542 | 0.596 | 0.763 |
| **MSE for E2F4** | 1.633 | 1.423 | 1.423 |

**Fig 5. Estimation of E2Fs accumulation over the cell cycle without marker identification with respect to nuclei segmentation.** Temporal evolution of E2F3A, E2F8 and E2F4 protein accumulation over the cell cycle in mouse intestinal epithelium, by considering positive/negative states for EdU and pH3 **a**, by only considering intensity for E2Fs, EdU and pH3 with a manual segmentation for nuclei **b**, with a nuclei segmentation obtained for the confocal images with a Mask R-CNN model trained with the widefield images and a manual segmentation for the widefield images **c**, and with a nuclei segmentation obtained for the widefield images with a Mask R-CNN model trained with the confocal images and a manual segmentation for the confocal images **d**.

## Discussion

This study demonstrates that the Mask R-CNN approach, coupled with transfer learning and data augmentation, can produce highly accurate nuclear segmentation of fluorescent images, even in a complex tissue such as the intestinal epithelium, by considering a small training dataset. Outstanding segmentation results are achieved for different imaging modalities, notably including widefield fluorescence microscopy, a ubiquitous imaging mode in cell biology research labs that produces challenging images of low sharpness and poor boundary definition. This is of major interest, as segmentation is a key task in image analysis of mammalian tissues where variability across tissue types is high and the number of modalities large and ever increasing. Therefore, the quest for comprehensive training datasets will be long and biology laboratories need efficient strategies to process their own images. In this context, the Inception-V3 approach also obtains satisfactory segmentation accuracy, but the computation time, especially for training, represents a major limitation. Though the U-Net approach has demonstrated its utility in many biomedical studies, it is unable to generalize the ability to segment nuclei when the images are acquired with different modalities.

Our results also attest to the superiority and efficiency of the deep learning approach for identifying the presence and state of nuclear markers. As the goal is to make a decision for each cell, the Inception-V3 architecture is well adapted to the task, computation time not being the issue that it is for nuclear segmentation. Considering pixel-based input patches improves performance if the training dataset is small. Moreover, transfer learning from nuclear segmentation improves marker identification. This is explained by the fact that image features that are suited for nuclear segmentation are also meaningful for identifying nuclear markers. This is an interesting observation, as the same strategy could be used in an n-steps pipeline analysis. Indeed, a deep learning classifier could be trained for the first analysis applied to a particular dataset, the resulting parameters used to initialize the classifier for the second analysis, and so on.

While deep learning approaches for both nuclear segmentation and nuclear marker identification show high accuracy, some level of discrepancy between desired and actual results is expected. Thus, it is important to allow the user to correct the results if higher accuracy is required. This need for interactive post-processing of the results motivated us to design and implement the ImageJ plugin Annotater. This plugin can be used to annotate data from scratch or to perform corrections. While Annotater could be used to obtain perfect accuracy even without prior automated segmentation and marker identification, starting from a good estimate markedly speeds up the process. For instance, delineating the boundaries of all nuclei in an image of the intestinal epithelium may take a trained user 1–1.5 hours whereas any user can correct the results obtained with the Mask R-CNN approach in less than five minutes.

We present a novel and powerful method to estimate temporal phenomena using quantitative analysis of 2D still images. Specifically, we showed how the time-variant concentrations of intranuclear proteins can be qualitatively obtained from snapshot images of cell populations of reasonable size (for example, FFPE images of hundreds of epithelial cells as we analyzed in this study). Compared to alternative methods like single-cell analysis of individual living cells, the method we put forward here potentially allows easier and less expensive experimental designs for many applications in which dynamic protein expression information is required. In this particular case, a smart initialization based on biologically relevant assumptions coupled with a global optimization approach enabled estimation of E2F accumulation over the cell cycle. We also demonstrate that this approach is robust to nuclei segmentation and does not require a marker identification to provide a good temporal estimation of the successive E2Fs waves of concentration.

## Materials and methods

### Mouse models

The E2f4$^{myc/myc}$ and E2f8$^{myc/myc}$ were generated using standard homologous recombination cloning techniques as described in [14]. E2f4$^{myc/myc}$, E2F8$^{myc/myc}$ and wild type controls were maintained on a mixed background (FVB/NT, 129v/Sv, C57BL/6NT). Tissues were collected from E13.5 intestinal epithelium.

### Ethics statement

Mouse usage and protocols were approved by the Institutional Animal Care and Use Committee at the Ohio State University and Medical University of South Carolina. Mice were housed under normal husbandry conditions (five or less animals per cage) in a vivarium with a 12-hour light/dark cycle.

### Immunostaining

Immunostaining was performed on a Bond Rx (Leica) or Ventana Discovery ULTRA (Roche) autostainer as per manufacturer's instructions as previously described [40, 41]. Primary antibodies and dilutions used in this study were as follows: pH3-S10 (Millipore; 06–570, 1:250), E2f3a (Millipore; 05–551, 1:100) and Myc-tag (Cell Signaling Technology; 2278, 1:100). EdU staining was performed following the manufacturer's protocol (Life Technologies; C10337).

### Image acquisition

Widefield micrographs were collected using a Nikon Eclipse Ni-U microscope with a DS-Qi2 camera and NIS-Elements Advanced Research software. Confocal micrographs were collected using the Olympus FV 1000 Filter Confocal system in the Campus Microscopy and Imaging Facility at the Ohio State University.

### Nuclei segmentation

The five deep learning approaches were coded in Python with the Python libraries numpy [42], tensorflow [43], PyTorch [44], keras [45], scipy [46] and scikit-image [47]. The code with the parameters used to train and process all experiments presented in this manuscript is available at https://github.com/tpecot/NucleiSegmentationAndMarkerIDentification/tree/master/. It was written with jupyter notebooks and widgets in order to be used by biologists. Video tutorials are also available on the same web page.

**Training dataset.** The training dataset for nuclei segmentation consisted of twelve confocal and forty-five widefield 512 x 512 images annotated by a moderately-skilled researcher. Four different training datasets were used to evaluate the deep learning approaches and are summarized in S3 Table For the Inception-V3 approach, training and validation datasets were pulled together to obtain the input patches. Ninety percent of image patches were used for training and ten percent of image patches were used for validation.

**U-Net.** As the U-Net approach estimates a class at each pixel, three classes were defined to allow separating nuclei as proposed in [19]: inner nuclei, nuclei contours and background. To facilitate nuclei separation, the nuclei contours in the training dataset were dilated [19]. To reduce memory usage and limit over-fitting, the imaging field for images in the training dataset was set to 256 x 256 by randomly cropping the 512 x 512 input images. These cropped images were then normalized with a 1–99 quantile. A root mean square prop was used to estimate the parameters of the deep convolutional neural network by minimizing a weighted cross entropy loss to handle class imbalance for 100 epochs. The weights associated with each class

were defined from the training dataset as their inverse proportion. A data augmentation to increase the training dataset by a factor of 100 was processed before normalization with the imgaug python library [48] and included flipping, rotation, shearing, blurring, sharpness and brightness modifications, noise addition, intensity inversion and contrast modifications.

**Inception-V3.** The same three classes were defined and the nuclei contours were dilated for the Inception-V3 approach as for the U-Net approach. The imaging field was set to 65 x 65 pixel image patches, large enough to include at least one nucleus as suggested in [19]. For each image patch, only the class of the central pixel was trained or processed. The training images were normalized by dividing the intensity at each pixel by the median intensity over the image and by subtracting at each pixel the average intensity of a 65 x 65 neighborhood around it. To obtain a balanced training dataset, the same number of input image patches were defined for each class. A stochastic gradient descent was used to estimate the parameters of the deep convolutional neural network by minimizing a categorical cross entropy for 15 epochs. A modification of the input image patches was processed after normalization with the imgaug python library [48] and included the same image transformations that were used for the U-Net approach.

**U-Net and Inception-V3 post-processing.** For both U-Net and Inception-V3 approaches, nuclei were obtained by subtracting the nuclei contours from the inner nuclei. Let us define $\mathbf{S}_c = \{S_c(x)\}_{x \in \Omega}$ where $S_c(x)$ denotes the nuclei contour score obtained with the U-Net or the Inception-V3 approach at pixel $x$, and $\Omega$ is the regular grid of pixels in the image. Similarly, we define $\mathbf{S}_n = \{S_n(x)\}_{x \in \Omega}$ the inner nuclei score at each pixel $x$. The nuclei component $\mathbf{N} = \{N(x)\}_{x \in \Omega}$ is defined as:

$$N(x) = S_n(x) - S_c(x). \tag{1}$$

This nuclei component is then thresholded to define a binary image $\mathbf{N}_b$:

$$N_b(x) = \begin{cases} 1 & \text{if } N(x) > 0, \\ 0 & \text{otherwise.} \end{cases} \tag{2}$$

The individual nuclei are defined as the connected components of the binary image $\mathbf{N}_b$. This leads to an under segmentation when contours between touching nuclei are not well estimated. Consequently, we propose to apply the watershed algorithm [5] on the nuclei component $\mathbf{N}$ and to take advantage of the nuclei contour score $\mathbf{S}_c$ obtained from the U-Net or Inception-V3 output to refine the segmentation. Let us define $\mathbf{W} = \{W(x)\}_{i \in \Omega}$ the result of the watershed algorithm applied to the inverse intensity of the nuclei component $\mathbf{N}$ prealably convolved with a Gaussian filter of kernel size 2.5 for U-Net and 3.5 for Inception-V3. From $\mathbf{W}$, a set of K new nuclei separations $\{\mathcal{S}(k)\}_{k=1,...,K}$ is obtained. A score $\mathbf{S}_w = \{S_w(k)\}_{k=1,...,K}$ is defined for each new separation $\mathcal{S}(k)$ as follows:

$$S_w(k) = \sum_{x \in \mathcal{S}(k)} S_c(x) - S_n(x). \tag{3}$$

All nuclei separations $\mathcal{S}(k)$ such that $S_w(k) > 0$ are used to separate the nuclei obtained as the connected components of the binary image $\mathbf{N}_b$.

**Mask R-CNN.** Version 2.1 of Mask R-CNN [31] was used in this study. The backbone network was defined as the Resnet-101 deep convolutional neural network [49]. We used the code in [7] to define the only class in this study, *i.e.* the nuclei. A data augmentation to increase the training dataset by a factor of 100 was processed before normalization with the imgaug python library [48] and included resizing, cropping, flipping, rotation, shearing, blurring, sharpness and brightness modifications, noise addition, intensity inversion and contrast

modifications. Transfer learning with fine-tuning from a network trained on the coco dataset [33] was also applied. In the first epoch, only the region proposal network, the classifier and mask heads were trained. The whole network was then trained for the next three epochs.

**Stardist.**   Version 0.7.1 of Stardist [8] was used in this study. The backbone network was defined as a U-Net [27]. 512 x 512 input images were normalized with a 1–99 quantile. The data augmentation proposed by the authors was used to increase the training dataset. The whole network was trained for 400 epochs.

**Cellpose.**   Version 0.6.5 of Cellpose [9] was used in this study. The backbone network was defined as a U-Net [27]. 512 x 512 input images were normalized with a 1–99 quantile. A data augmentation to increase the training dataset was processed before normalization with the imgaug python library [48] and included flipping, rotation, shearing, blurring, sharpness and brightness modifications, noise addition, intensity inversion and contrast modifications. The whole network was trained for 400 epochs.

A summary of all the different steps used for each deep learning approach is available in S2 Table. The time computation for training and processing each individual method is shown in S1 and S4 Tables.

**Evaluation.**   As proposed by Caicedo *et al.* [32], we used the F1 score with respect to the Intersection over Union (*IoU*) to compare the three deep learning approaches. We evaluated three different images for both confocal and widefield modalities annotated by a pathologist (a high-skilled researcher). Let $\mathbf{O}_{GT} = \{O_{GT}(e)\}_{e=1,\ldots,n}$ be the set of $n$ ground truth nuclei, as defined by the pathologist, and $\mathbf{O}_E = \{O_E(e)\}_{e=1,\ldots,m}$ be the set of $m$ estimated nuclei with a deep learning approach. The *IoU* defined between the truth nucleus $O_{GT}(e_1)$ and the estimated nucleus $O_E(e_2)$ was defined as:

$$IoU(O_{GT}(e_1), O_E(e_2)) = \frac{O_{GT}(e_1) \cap O_E(e_2)}{O_{GT}(e_1) \cup O_E(e_2)}. \tag{4}$$

An $IoU(O_{GT}(e_1), O_E(e_2))$ equal to 0 implies that $O_{GT}(e_1)$ and $O_E(e_2)$ do not share any pixel while an $IoU(O_{GT}(e_1), O_E(e_2))$ equal to 1 means that $O_{GT}(e_1)$ and $O_E(e_2)$ are identical. An $IoU(O_{GT}(e_1), O_E(e_2))$ equal to 0.5 ensures that $IoU(O_{GT}(e_1), O_E(e_3)) < 0.5, \forall e_3 \neq e_2$ as a nucleus cannot share half of its area with more than one nucleus. Consequently, the F1 score for a given *IoU* threshold $t > 0.5$ can be defined as:

$$F1(t) = \frac{2 \times TP(t)}{2 \times TP(t) + FN(t) + FP(t)}, \tag{5}$$

where

$$TP(t) = \sum_{e_1 \in \{1,\ldots,n\}, e_2 \in \{1,\ldots,m\}} \mathbb{1}(IoU(O_{GT}(e_1), O_E(e_2)) > t), \tag{6}$$

$$FN(t) = \sum_{e_1 \in \{1,\ldots,n\}} \mathbb{1}(IoU(O_{GT}(e_1), O_E(e_2)) < t), \forall e_2 \in \{1,\ldots,m\}, \tag{7}$$

$$FP(t) = \sum_{e_2 \in \{1,\ldots,m\}} \mathbb{1}(IoU(O_{GT}(e_1), O_E(e_2)) < t), \forall e_1 \in \{1,\ldots,n\}, \tag{8}$$

and

$$\mathbb{1}(\mathcal{C}) = \begin{cases} 1 \text{ if } \mathcal{C} \text{ is true,} \\ \\ 0 \text{ otherwise.} \end{cases} \tag{9}$$

With a threshold $t = 0.5$, this metric gives the accuracy of a method to identify the correct number of nuclei, while with thresholds in the range $0.5 - 1$, it evaluates the localization accuracy of the identified nuclear contours.

## Marker identification

### Training dataset

As Mask R-CNN and U-Net are not suited for the task, the Inception-V3 method was used for marker identification. The training dataset consisted of eight confocal images for E2F3A, five confocal images for E2F8, four confocal images for EdU, three confocal images for pH3, twenty-one widefield images for E2F3A and fifteen widefield images for E2F8, E2F4, EdU and pH3, all annotated by a moderately-skilled researcher. Ninety percent of image patches were used for training and ten percent for validation. Two training strategies were designed to define the input image patches: The first only considered image patches centered at each nucleus center, while the second strategy included image patches centered at each pixel in the segmented nuclei, drastically increasing the amount of training data.

### Inception-V3

Two classes (positive and negative) were defined for the E2Fs, while three classes (diffuse, punctate and negative) were defined for EdU and pH3. As for nuclei segmentation, the imaging field was set to 65 x 65 pixel image patches to ensure inclusion of at least one entire nucleus. The training image patches were normalized by subtracting at each pixel the average intensity in a 65 x 65 neighborhood. Intensities were not divided by the median intensity over the image as they were for nuclei segmentation because the median can be equal to zero for sparse markers such as pH3. To obtain a balanced training dataset, the same number of input image patches was defined for each class. A stochastic gradient descent was used to estimate the parameters of the deep convolutional neural network by minimizing a categorical cross entropy for 10 epochs. To increase the size of the nuclei-based training dataset 100-fold, a data augmentation that included flipping, rotation, shearing, blurring, sharpness and brightness modifications, noise addition, inversion and contrast modifications was performed after normalization with the imgaug python library [48]. A transfer learning with fine-tuning from the Inception-V3 network trained for nuclei segmentation on confocal and widefield images was performed on the pixel-based training dataset. In the first epoch, only the last layer of the network was trained. The whole network was then trained for the next five epochs.

### Thresholding

Marker identification was also evaluated when thresholding marker intensity in the nuclei with the ImageJ plugin Annotater. For E2Fs, nuclei with more than thirty-five percent of their area above a manually defined threshold were considered positive. For EdU, nuclei with more that seventy percent of their area above a manually defined threshold were considered diffuse, while nuclei with more than fifteen percent but less than seventy percent of their area above the same threshold were considered punctate. For pH3, nuclei with more that ninety percent of their area above a manually defined threshold were considered diffuse, while nuclei with more than ten percent but less than ninety percent of their area above the same threshold were considered punctate.

### Evaluation

Evaluation was performed using three confocal and three widefield images for E2F3A and E2F8, using three widefield images for E2F4, and using two confocal and three widefield

images for EdU and pH3. All the images used for evaluation were annotated by a moderately-skilled researcher. The nuclei locations were known and only their class (positive/negative or diffuse/punctate/negative) was evaluated. Let $\mathbf{C}_{GT} = \{C_{GT}(e)\}_{e=1,\ldots,n}$ be the ground truth class for the $n$ nuclei and let $\mathbf{C}_E = \{C_E(e)\}_{e=1,\ldots,n}$ be the estimated class for each nucleus. Class estimation accuracy is then defined as:

$$\text{accuracy} = \frac{TP}{n}, \tag{10}$$

where

$$TP = \sum_{e=1}^{n} \mathbb{1}(C_{GT}(e) = C_E(e)). \tag{11}$$

## ImageJ plugin Annotater

We developed the plugin entitled *Annotater* to draw nuclear contours manually to define training datasets and to interactively correct deep learning-based nuclei segmentations. This plugin also provides for manual characterization of markers associated with nuclei and a thresholding tool for specifying an intensity threshold and the proportion of pixels in a nucleus above that threshold that defines a positive nucleus. Finally, Annotater can generate images in which segmented nuclei and marker characterization are overlaid on the input image and can extract nuclear features as defined in the *Set Measurements* window in ImageJ. The java code of the plugin is available at https://github.com/tpecot/Annotater/tree/master/src/main/java/edu/musc/tsl and the plugin is available at https://github.com/tpecot/Annotater. Video tutorials showing how to use the plugin are available at https://github.com/tpecot/Annotater.

## E2Fs accumulation over the cell cycle from 2D still images

**Data.** As fluorescence intensity is proportional to protein concentration [36, 37], E2Fs nuclear accumulation over the cell cycle can be estimated from the intensity levels observed in the nuclei. To minimize the effects of noise while still being able to observe fluorescence fluctuations, we decided to define four levels of intensity, which is widely used in both diagnostic pathology and biomedical research [38, 39]. In the previous section about marker identification, positive and negative cells were manually selected to define the training and evaluation datasets. By definition, negative cells do not express proteins and were consequently assigned an intensity equal to 0. Then, for each individual image, the average fluorescence intensity for each positive cell was processed and the range of average intensities from the lowest to the highest was binned into three to define levels 1 to 3. Defining these levels from the range of average intensity observed in positive nuclei amounts to normalizing intensity over the images. S7a and S7b Fig shows the E2F8 channel for a confocal and a widefield image where the positive E2F8 nuclei are overlaid as orange circles while negative E2F8 nuclei are overlaid as red circles. The corresponding histograms for the average E2F8 intensity observed in nuclei are shown in S7c and S7d Fig. The colors in these histograms correspond to the E2F8 levels used to estimate the concentration evolution over the cell cycle. The intensity minimum and maximum for each level are different for the two images as they depend on the lowest and highest nuclear average intensities, which are different from one image to the other. However, for a given image, the level width is the same for levels 1 to 3. Additionally, some E2F8 negative nuclei show an average nuclear intensity similar to the one observed in nuclei with an intensity level of 1 (bars that are blue and yellow). This happens for nuclei with non-specific E2F8

intensity. On the other hand, the state of EdU and pH3 markers for each nucleus was already known from the marker identification as being diffuse, punctate or negative.

More formally, let us define $I_e^{\text{Ch1}} \in \{0, 1, 2, 3\}$ the average intensity in Ch1 for nucleus $e \in \{1, \ldots, n\}$ in an image with $n$ nuclei, where Ch1 is a channel corresponding to E2F3A, E2F8 or E2F4. Similarly, $I_e^{\text{Ch2}} \in \{0, 1, 2, 3\}$ is the average intensity in Ch2 for nucleus $e$ if Ch2 is E2F4 or E2F8 and $I_e^{\text{Ch2}} \in \{0, \text{diffuse}, \text{punctate}\}$ is the state of nucleus $e$ if Ch2 is EdU or pH3. The estimation of fluorescence accumulation over the cell cycle is performed in 3 steps: i) initialization, ii) registration with respect to cell cycle, iii) global optimization.

**Initialization.**   Let us define $h(I^{\text{Ch1}} = u_1, I^{\text{Ch2}} = u_2)$ the number of nuclei for which the intensity in Ch1 is equal to $u_1$ and the intensity or state in Ch2 is equal to $u_2$:

$$h(I^{\text{Ch1}} = u_1, I^{\text{Ch2}} = u_2) = \sum_{e=1}^{n} \mathbb{1}(I_e^{\text{Ch1}} = u_1)\mathbb{1}(I_e^{\text{Ch2}} = u_2). \tag{12}$$

We can now define $\mathbf{H} = \{H(I^{\text{Ch1}} = u_1, I^{\text{Ch2}} = u_2)\}_{u_1 = \{0,1,2,3\}, u_2 = \{0,1,2,3\} \text{ or } u_2 = \{0, \text{diffuse}, \text{punctate}\}}$ the 2D intensity histogram as follows:

$$H(I^{\text{Ch1}} = u_1, I^{\text{Ch2}} = u_2) = \frac{h(I^{\text{Ch1}} = u_1, I^{\text{Ch2}} = u_2)}{n}. \tag{13}$$

An example of a 2D histogram of E2F3A intensity and pH3 patterns is shown in S8a Fig. According to our first and second assumptions, *i.e.* time depends on the proportion of cells in the images and temporal evolution of protein accumulation is similar in all observed cells, 2D intensity histograms provide the proportion of nuclei for combination of E2Fs intensities or E2Fs intensity and EdU/pH3 patterns. For example, in the image considered to build the histogram shown in S8a Fig, thirty-four percent of nuclei have an intensity of 2 for E2F3A and are negative for pH3. Combining these proportions allows to define the evolution of the intensity/ state of the channels over the cell cycle. We propose to initialize the evolution of fluorescence over the cell cycle through a graph optimization procedure. Let us define a graph $\mathcal{G}(\mathcal{E}, \mathcal{V})$ with $|\mathcal{V}|$ vertices corresponding to the different non-zero proportions of markers from a 2D histogram and $|\mathcal{E}|$ edges between neighbor vertices in the same histogram. For instance, in S8a Fig, there are nine vertices $\{\mathcal{V}_{E2F3A=0,pH3=\text{negative}}, \mathcal{V}_{E2F3A=1,pH3=\text{negative}}, \ldots, \mathcal{V}_{E2F3A=0,pH3=\text{diffuse}}\}$ represented as circles and twenty-two edges $\{\mathcal{E}(\mathcal{V}_{E2F3A=0,pH3=\text{negative}} \rightarrow \mathcal{V}_{E2F3A=1,pH3=\text{negative}}), \ldots, \mathcal{E}(\mathcal{V}_{E2F3A=0,pH3=\text{diffuse}} \rightarrow \mathcal{V}_{E2F3A=0,pH3=\text{punctate}})\}$ represented as unidirectional arrows. A cost $\mathcal{P}(\mathcal{E}(\mathcal{V}_{v_1}, \mathcal{V}_{v_2}))$ is then assigned to each edge. From [14], we know that EdU is diffuse during the first half of S phase and then punctate during the second half of S phase. It implies that:

$$\begin{aligned} \mathcal{P}(\mathcal{E}(\mathcal{V}_{G=u,EdU=\text{negative}}, \mathcal{V}_{G=u,EdU=\text{diffuse}})) &= 0, \\ \mathcal{P}(\mathcal{E}(\mathcal{V}_{G=u,EdU=\text{diffuse}}, \mathcal{V}_{G=u,EdU=\text{punctate}})) &= 0, \\ \mathcal{P}(\mathcal{E}(\mathcal{V}_{G=u,EdU=\text{punctate}}, \mathcal{V}_{G=u,EdU=\text{negative}})) &= 0, \end{aligned} \tag{14}$$

while

$$\begin{aligned} \mathcal{P}(\mathcal{E}(\mathcal{V}_{G=u,EdU=\text{diffuse}}, \mathcal{V}_{G=u,EdU=\text{negative}})) &= \infty, \\ \mathcal{P}(\mathcal{E}(\mathcal{V}_{G=u,EdU=\text{punctate}}, \mathcal{V}_{G=u,EdU=\text{diffuse}})) &= \infty, \\ \mathcal{P}(\mathcal{E}(\mathcal{V}_{G=u,EdU=\text{negative}}, \mathcal{V}_{G=u,EdU=\text{punctate}})) &= \infty, \end{aligned} \tag{15}$$

where $G \in \{E2F3A, E2F8, E2F4\}$ and $u \in \{0, 1, 2, 3\}$. From [14], we also know that pH3 is first

punctate during the second half of S phase and G2 and then diffuse during mitosis, so

$$
\begin{aligned}
\mathcal{P}(\mathcal{E}(\mathcal{V}_{G=u,pH3=\text{negative}}, \mathcal{V}_{G=u,pH3=\text{punctate}})) &= 0, \\
\mathcal{P}(\mathcal{E}(\mathcal{V}_{G=u,pH3=\text{punctate}}, \mathcal{V}_{G=u,pH3=\text{diffuse}})) &= 0, \\
\mathcal{P}(\mathcal{E}(\mathcal{V}_{G=u,pH3=\text{diffuse}}, \mathcal{V}_{G=u,pH3=\text{negative}})) &= 0, \\
\mathcal{P}(\mathcal{E}(\mathcal{V}_{G=u,pH3=\text{punctate}}, \mathcal{V}_{G=u,pH3=\text{negative}})) &= \infty, \\
\mathcal{P}(\mathcal{E}(\mathcal{V}_{G=u,pH3=\text{diffuse}}, \mathcal{V}_{G=u,pH3=\text{punctate}})) &= \infty, \\
\mathcal{P}(\mathcal{E}(\mathcal{V}_{G=u,pH3=\text{negative}}, \mathcal{V}_{G=u,pH3=\text{diffuse}})) &= \infty,
\end{aligned}
\tag{16}
$$

where $G \in \{E2F3A, E2F8, E2F4\}$ and $u \in \{0, 1, 2, 3\}$. Finally, the evolution of E2Fs intensity needs to verify our third assumption, *i.e.* concentrations of E2Fs are concave downward parabolas, which explains why edges exist only between neighboring vertices in the 2D histograms:

$$
\mathcal{P}(\mathcal{E}(\mathcal{V}_{G=u_1,M=s}, \mathcal{V}_{G=u_2,M=s})) = \begin{cases} 0 \text{ if } |u_1 - u_2| = 1, \\ \\ \infty \text{ otherwise,} \end{cases}
\tag{17}
$$

where $G \in \{E2F3A, E2F8, E2F4\}$, $u_1 \in \{0, 1, 2, 3\}$, $u_2 \in \{0, 1, 2, 3\}$, $M \in \{\text{EdU}, \text{pH3}\}$ and $s \in \{\text{negative, punctate, diffuse}\}$. The evolution of the fluorescence is initialized as the sequence of edges that goes through all vertices with a minimum cost. For instance, the only sequence of edges with a cost equal to 0 in the 2D histogram shown in S8a Fig is
$\{\mathcal{V}_{E2F3A=0,pH3=\text{negative}}, \mathcal{V}_{E2F3A=1,pH3=\text{negative}}, \mathcal{V}_{E2F3A=2,pH3=\text{negative}}, \mathcal{V}_{E2F3A=3,pH3=\text{negative}},$
$\mathcal{V}_{E2F3A=3,pH3=\text{punctate}}, \mathcal{V}_{E2F3A=2,pH3=\text{punctate}}, \mathcal{V}_{E2F3A=1,pH3=\text{punctate}}, \mathcal{V}_{E2F3A=0,pH3=\text{punctate}}, \mathcal{V}_{E2F3A=0,pH3=\text{diffuse}}\}$.
This sequence corresponds to the evolution of E2F3A and pH3 shown in S8b Fig.

We know that E2F4 shows two waves of expression over the cell cycle [14], but a sequence of edges over a 2D histogram cannot reflect two downward parabolas. Consequently, splitting vertex(ices) is necessary. A cost of 1 is associated to each vertex splitting and added to the total cost. The split vertices give rise to multiple vertices containing equal proportion. For combinations of E2Fs with EdU as well as combinations of two different E2Fs, vertex splitting can also be required and may lead to E2Fs evolutions that do not look exactly like downward parabolas. This phenomenon arises from slight errors in nuclear intensity measurements, most likely due to image noise. A 2D histogram for the combination of E2F3A and E2F4 is shown in S9a Fig, the sequence of edges over this histogram for which vertices were split is shown in S9b Fig and the corresponding evolution of E2F3A and E2F4 is shown in S9c Fig.

By applying this cost minimization to all images, initializations for the eight different combinations of markers and averaged for each mouse are obtained. These initializations are shown in S10 Fig. From these initializations, eight couples of marker sequences are defined:

$$
\begin{aligned}
&\left\{ {}^{\text{init}}X_i^{EdU-E2F3A}(c), {}^{\text{init}}X_i^{E2F3A-EdU}(c) \right\}_{c=1,\dots,B}, \\
&\left\{ {}^{\text{init}}X_i^{pH3-E2F3A}(c), {}^{\text{init}}X_i^{E2F3A-pH3}(c) \right\}_{c=1,\dots,B}, \\
&\left\{ {}^{\text{init}}X_i^{EdU-E2F8}(c), {}^{\text{init}}X_i^{E2F8-EdU}(c) \right\}_{c=1,\dots,B}, \\
&\left\{ {}^{\text{init}}X_i^{pH3-E2F8}(c), {}^{\text{init}}X_i^{E2F8-pH3}(c) \right\}_{c=1,\dots,B}, \\
&\left\{ {}^{\text{init}}X_i^{EdU-E2F4}(c), {}^{\text{init}}X_i^{E2F4-EdU}(c) \right\}_{c=1,\dots,B}, \\
&\left\{ {}^{\text{init}}X_i^{pH3-E2F4}(c), {}^{\text{init}}X_i^{E2F4-pH3}(c) \right\}_{c=1,\dots,B}, \\
&\left\{ {}^{\text{init}}X_i^{E2F3A-E2F8}(c), {}^{\text{init}}X_i^{E2F8-E2F3A}(c) \right\}_{c=1,\dots,B}, \\
&\left\{ {}^{\text{init}}X_i^{E2F3A-E2F4}(c), {}^{\text{init}}X_i^{E2F4-E2F3A}(c) \right\}_{c=1,\dots,B},
\end{aligned}
\tag{18}
$$

where $^{\mathrm{init}}X_i^{M1-M2}(c)$ provides the level of fluorescence or pattern for marker $M1 \in \{E2F3A,$ $E2F8, E2F4, EdU, pH3\}$ when combined with marker $M2 \in \{E2F3A, E2F8, E2F4, EdU, pH3\}$ for mouse $i$ at time $c$, $M1 \neq M2$, and $B$ is the number of time bins used to quantize the cell cycle: 100 bins in practice. The goal is to estimate the sequence for each of these couples of combinations.

**Registration.** As EdU and pH3 expression patterns with respect to the cell cycle are known, E2Fs are first registered with respect to these two markers. We know that pH3 is diffuse during mitosis. Consequently, this pattern is considered as the reference and each initialization of any combination with pH3 ends with a diffuse pH3 pattern corresponding to the end of the cell cycle. This implies that the pH3 sequences are known and defined as $\mathbf{X}_i^{G-pH3}$ for each mouse $i$. A circular permutation is operated next to estimate the $X_i^{EdU-G}$ where $G \in \{E2F3A,$ $E2F8, E2F4\}$ as follows:

$$\min_{\{^{0-}\mathbf{X}_i^{EdU-G}, ^{0-}\mathbf{X}_i^{G-EdU}\}} \sum_{j=1}^{N_G} \|^{0-}\mathbf{X}_i^{G-EdU} - \mathbf{X}_j^{G-pH3}\|_2^2,$$

$$\text{subject to} \quad ^{0-}X_i^{G-EdU}(c) = ^{\mathrm{init}}X_i^{G-EdU}(c-q), q \in \{1, \ldots, B\},$$

$$\forall i \in \{1, \ldots, N_G\}, \forall G \in \{E2F3A, E2F8, E2F4\}, \tag{19}$$

where $N_G = 3$ for $E2F3A$ and $E2F8$, $N_G = 2$ for $E2F4$, and $^{0-}X_i^{G-EdU}(c-q) = ^{\mathrm{init}}X_i^{G-EdU}(B-(c-q))$ if $(c-q)$ is negative. We solve this optimization problem by computing $\sum_{j=1}^{N_G} \|^{0-}\mathbf{X}_i^{G-EdU} - \mathbf{X}_j^{G-pH3}\|_2^2$ for all possible shifts $c = \{1, \ldots, B\}$. The solution corresponds to the shift $c$ that gives the minimum sum. To better estimate EdU and initialize E2Fs, we now use the E2Fs expression:

$$\min_{\{^0\mathbf{X}_i^{EdU-G1}, ^0\mathbf{X}_i^{G1-EdU}\}} \sum_{j=1}^{N_{G1}} \|^0\mathbf{X}_i^{G1-EdU} - \mathbf{X}_j^{G1-pH3}\|_2^2 +$$

$$\sum_{j=1}^{N_{G2}} \|^0\mathbf{X}_i^{EdU-G1} - ^0\mathbf{X}_j^{EdU-G2}\|_2^2 +$$

$$\sum_{j=1}^{N_{G3}} \|^0\mathbf{X}_i^{EdU-G1} - ^0\mathbf{X}_j^{EdU-G3}\|_2^2,$$

$$\text{subject to} \quad ^0X_i^{EdU-G1}(c) = ^{0-}X_i^{EdU-G1}(c-q), q \in \{1, \ldots, B\}, \forall i \in \{1, \ldots, N_{G_1}\}$$

where $\{G1, G2, G3\}$ are successively $\{E2F3A, E2F8, E2F4\}$, $\{E2F8, E2F3A, E2F4\}$ and $\{E2F4,$ $E2F8, E2F3A\}$, $N_{Gi} = 3$ if $Gi$ is $E2F3A$ or $E2F8$ and $N_{Gi} = 2$ if $Gi$ is $E2F4$. Again, we solve this optimization problem by computing the distances (20) for all possible shifts $c = \{1, \ldots, B\}$ and obtain the solution with the minimum value. Now, the temporal evolutions of EdU $\mathbf{X}_i^{EdU-G}$ and pH3 $\mathbf{X}_i^{pH3-G}$ are known for each mouse $i$ and each genotype $G \in \{E2F3A, E2F8, E2F4\}$. This implies that EdU and pH3 evolutions will not change but can be permuted for a given state of EdU or pH3. The E2Fs concentrations and patterns for EdU and pH3 after registration are shown in S11 Fig.

**Assignment problem.** Now that EdU and pH3 are known, we define the estimation of E2Fs as an assignment problem where the goal is to find the permutations for each $\mathbf{X}_i^{G-M}$, $G \in$ $\{E2F3A, E2F8, E2F4\}$ and $M \in \{EdU, pH3\}$ such that E2Fs accumulation over time are the most similar across data while EdU and pH3 are already known. We model this estimation as

an assignment problem and define the following square cost matrices:

$$
Cost({}^{(z)}X_i^{G1-M1}(c_1, c_2)) = \sum_{j=1}^{N_{G1}} \|{}^{(z-1)}X_i^{G1-M1}(c_1) - {}^{(z)}X_j^{G1-M2}(c_2)\|_2^2 +
$$

$$
\sum_{j=1}^{N_{G2}} \|{}^{(z-1)}X_i^{G1-M1}(c_1) - {}^{(z)}X_j^{G1-G2}(c_2)\|_2^2 +
$$

$$
\sum_{j=1}^{N_{G3}} \|{}^{(z-1)}X_i^{G1-M1}(c_1) - {}^{(z)}X_j^{G1-G3}(c_2)\|_2^2 + \qquad (20)
$$

$$
\|{}^{(z)}X_i^{M1-G1}(c_1) - X_i^{M1-G1}(c_2)\|_\infty +
$$

$$
\frac{\sqrt{(c_1-c_2)^2}}{z}, \forall i \in \{1, \ldots, N_{G1}\},
$$

where the $\ell_\infty$ norm is used to ensure that EdU and pH3 states stay the same over the cell cycle, $\{G1, G2, G3\}$ are successively $\{E2F3A, E2F8, E2F4\}$, $\{E2F8, E2F3A, E2F4\}$ and $\{E2F4, E2F8, E2F3A\}$, $N_{Gi} = 3$ if $Gi$ is $E2F3A$ or $E2F8$ and $N_{Gi} = 2$ if $Gi$ is $E2F4$, $\{M1, M2\}$ are successively $\{EdU, pH3\}$ and $\{pH3, EdU\}$, $z$ corresponds to the iteration of the algorithm and:

$$
Cost({}^{(z)}X_i^{E2F3A-G1}(c_1, c_2))
$$

$$
= \sum_{j=1}^{3} \{\|{}^{(z-1)}X_i^{E2F3A-G1}(c_1) - {}^{(z-1)}X_j^{E2F3A-EdU}(c_2)\|_2^2 +
$$

$$
\|{}^{(z-1)}X_i^{E2F3A-G1}(c_1) - {}^{(z-1)}X_j^{E2F3A-pH3}(c_2)\|_2^2\} +
$$

$$
\sum_{j=1}^{N_{G1}} \{\|{}^{(z-1)}X_i^{G1-E2F3A}(c_1) - {}^{(z-1)}X_j^{G1-EdU}(c_2)\|_2^2 + \qquad (21)
$$

$$
\|{}^{(z-1)}X_i^{G1-E2F3A}(c_1) - {}^{(z-1)}X_j^{G1-pH3}(c_2)\|_2^2\} +
$$

$$
\sum_{j=1}^{N_{G2}} \|{}^{(z-1)}X_i^{E2F3A-G1}(c_1) - {}^{(z-1)}X_j^{E2F3A-G2}(c_2)\|_2^2 +
$$

$$
\frac{\sqrt{(c_1-c_2)^2}}{z}, \forall i \in \{1, \ldots, N_{G1}\},
$$

where $\{G1, G2\}$ are successively $\{E2F8, E2F4\}$ and $\{E2F4, E2F8\}$, $N_{Gi} = 3$ if $Gi$ is $E2F8$ and $N_{Gi} = 2$ if $Gi$ is $E2F4$. Each one of these five minimizations is processed one after another for iterations $z = \{1, \ldots, B\}$ by using the Hungarian algorithm [24, 25]. In Eq (20), the three first lines reflect the goal to have a given E2F accumulation over the cell cycle as similar as possible when combined with EdU, pH3 or another E2F; the fourth line corresponds to the constraint that EdU and pH3 are known and their state over the cell cycle cannot change, even though there can be permutations for a given state; and the last line defines a local constraint that forces permutations to be restricted in time for the first iterations and allows them to grow when iterations get larger. The latter constraint is used because the data is well initialized so the final estimation should not permute points directly all over the cell cycle. Eq (21) takes into account the differences between E2F3A accumulation over the cell cycle when combined with any marker or any other E2F, with the same constraint as in Eq (20). Finally, a spline fitting is applied to each of the E2Fs across all experiments to get a final accumulation over the cell cycle, which is shown in Fig 3. The code with the parameters used to estimate E2Fs

accumulation over the cell cycle after initialization is available at https://github.com/tpecot/EstimationOfProteinConcentrationOverTime.

**Simulated data.** We generated simulations to evaluate the accuracy of the temporal estimation of protein concentration. For each simulation, the following concentrations are generated:

$$
\begin{aligned}
\mathbf{y}_i^{\text{E2F3A}_j} &\in \{0, \ldots, \text{nb}_\text{I}\}, i \in \{0, \ldots, \text{nb}_\text{T}\}, j \in \{0, \ldots, \text{nb}_\text{S}\}, \\
\mathbf{y}_i^{\text{E2F8}_j} &\in \{0, \ldots, \text{nb}_\text{I}\}, i \in \{0, \ldots, \text{nb}_\text{T}\}, j \in \{0, \ldots, \text{nb}_\text{S}\}, \\
\mathbf{y}_i^{\text{E2F4}_j} &\in \{0, \ldots, \text{nb}_\text{I}\}, i \in \{0, \ldots, \text{nb}_\text{T}\}, j \in \{0, \ldots, \text{nb}_{\text{S}_2}\}, \\
\mathbf{y}_i^{\text{EdU}_j} &\in \{0, \text{diffuse}, \text{punctate}\}, i \in \{0, \ldots, \text{nb}_\text{T}\}, j \in \{0, \ldots, \text{nb}_\text{S}\}, \\
\mathbf{y}_i^{\text{pH3}_j} &\in \{0, \text{punctate}, \text{diffuse}\}, i \in \{0, \ldots, \text{nb}_\text{T}\}, j \in \{0, \ldots, \text{nb}_\text{S}\},
\end{aligned}
\tag{22}
$$

where $\text{nb}_\text{I}$ is the number of bins for intensity, $\text{nb}_\text{T}$ is the number of bins for time and $\text{nb}_\text{S}$ is the number of samples for E2F3A, E2F8, EdU and pH3 while $\text{nb}_{\text{S}_2}$ is the number of samples for E2F4, corresponding to mice in our study. $\mathbf{Y}^{\text{E2F3A}}$ and $\mathbf{Y}^{\text{E2F8}}$ are simulated as concentrations starting at intensity 0 to $\text{nb}_\text{I}$, and then back to 0 with a duration for each intensity bin randomly generated between 1 and $\frac{2}{3*\text{nb}_\text{I}} * \text{nb}_\text{T}$ to ensure that the concentration is shorter that the simulation. The starting point for these concentrations is also randomly generated. $\mathbf{Y}^{\text{E2F4}}$ is simulated as two waves of concentrations starting at intensity 0 to $\text{nb}_\text{I}$, and then back to 0 with a duration for each intensity bin randomly generated between 1 and $\frac{1}{3*\text{nb}_\text{I}} * \text{nb}_\text{T}$ to ensure that both waves can fit in the simulation duration. The starting point for both waves are randomly generated, with the constraint that the second wave starts after the first one ends. $\mathbf{Y}^{\text{EdU}}$ and $\mathbf{Y}^{\text{pH3}}$ are simulated as successive diffuse and punctate patterns for EdU and successive punctate and diffuse patterns for pH3, with a duration for each pattern randomly generated between 1 and $\frac{2}{3*\text{nb}_\text{I}} * \text{nb}_\text{T}$. The starting point for these markers is also randomly generated, with the constraint that pH3 starts after EdU. To evaluate the robustness of our approach, noise was randomly added to affect a range from 0% to 50% of corrupted time bins. In addition, five different simulations were randomly generated for each set of parameters. The performance was evaluated by measuring the mean squared error (MSE) between the generated and estimated concentrations of E2Fs, EdU and pH3.

Three different scenarios were designed to evaluate the performance of our approach. In a first set of experiments, the number of samples was evaluated and defined as $\{n_1 = 1; n_2 = 1\}$, $\{n_1 = 2; n_2 = 2\}$, $\{n_1 = 3; n_2 = 2\}$, $\{n_1 = 3; n_2 = 3\}$ and $\{n_1 = 5; n_2 = 5\}$, with $\text{nb}_\text{I} = 4$ and $\text{nb}_\text{T} = 100$. The results are shown in S12 Fig. Then, the influence of the number of time bins was estimated by considering $\text{nb}_\text{T} = \{20, 50, 75, 100, 125, 150, 200\}$ with $\{n_1 = 3; n_2 = 2\}$ and $\text{nb}_\text{I} = 4$. The results are shown in S13 Fig. Finally, the impact of the number of intensity bins was evaluated by defining $\text{nb}_\text{I} = \{2, 3, 4, 5, 10, 20\}$ with $\{n_1 = 3; n_2 = 2\}$ and $\text{nb}_\text{T} = 100$. The results are shown in S14 Fig.

## Supporting information

**S1 Fig. Example images.** One example for each combination of markers and modality for fluorescence images used in the study. The combination of markers, number of images of each type, modality, and number of mice used in the study are shown on top of the images. Note the small size of the training set. Scale bar = 20μm.
(TIF)

**S2 Fig. Evaluation of U-Net for nuclei segmentation.** F1 score for multiple IoU thresholds obtained with the U-Net approach by trainig on confocal, half or all widefield images, and on confocal and widefield images without data augmentation or post-processing **a-b**, with data augmentation **c-d**, with data augmentation and corrected watershed postprocessing **e-f**. The lines correspond to the average F1 score while the areas represent the standard error.
(TIF)

**S3 Fig. Evaluation of Inception-V3 for nuclear segmentation.** F1 score for multiple IoU thresholds obtained with the Inception-V3 approach by trainig on confocal, half or all widefield images, and on confocal and widefield images without data augmentation or post-processing **a-b**, with data augmentation **c-d**, with data augmentation and corrected watershed postprocessing **e-f**. The lines correspond to the average F1 score while the areas represent the standard error.
(TIF)

**S4 Fig. Evaluation of Mask R-CNN for nuclear segmentation.** F1 score for multiple IoU thresholds obtained with the Mask R-CNN approach by trainig on confocal, half or all widefield images, and on confocal and widefield images without data augmentation but with transfer learning **a-b**, without transfer learning but with data augmentation **c-d**, with data augmentation and transfer learning **e-f**. The lines correspond to the average F1 score while the areas represent the standard error.
(TIF)

**S5 Fig. Evaluation of Stardist for nuclear segmentation.** F1 score for multiple IoU thresholds obtained with the Stardist approach by trainig on confocal, half or all widefield images, and on confocal and widefield images without data augmentation **a-b** and with data augmentation **c-d**. The lines correspond to the average F1 score while the areas represent the standard error.
(TIF)

**S6 Fig. Evaluation of Cellpose for nuclear segmentation.** F1 score for multiple IoU thresholds obtained with the Cellpose approach by trainig on confocal, half or all widefield images, and on confocal and widefield images without data augmentation **a-b** and with data augmentation **c-d**. The lines correspond to the average F1 score while the areas represent the standard error.
(TIF)

**S7 Fig. Examples of intensity binning for E2F8. a-b** E2F8 channel of a confocal **a** and a widefield **b** image. E2F8 positive nuclei are overlaid as orange circles while E2F8 negative nuclei are overlaid as red circles. Scale bar = 20μm. **c-d** Histograms of the E2F8 intensity observed in the nuclei shown in images **a** (**c**) and **b** (**d**). The 3 levels of intensity used for concentration estimation are displayed with different colors.
(TIF)

**S8 Fig. Initialization of E2F3A and pH3 over time. a** 2D histogram of E2F3A intensity and pH3 patterns. **b** Initialization of E2F3A and pH3 over time from the 2D histogram shown in **a**.
(TIF)

**S9 Fig. Initialization of E2F3A and E2F4 over time. a** 2D histogram of E2F3A and E2F4 intensity. **b** Modified 2D histogram of E2F3A and E2F4 intensity with vertex splitting and set of edges giving the minimum cost. **c** Initialization of E2F3A and E2F4 over time from the set of edges shown in **b**.
(TIF)

**S10 Fig. Initialization of E2Fs with respect to EdU and pH3. a-b** Initialization of E2F3A with respect to **a** EdU (n = 3) and **b** pH3 (n = 3). **c-d** Initialization of E2F8 concentration with respect to **c** EdU (n = 3) and **d** pH3 (n = 3). **e-f** Initialization of E2F4 concentration with respect to **e** EdU (n = 2) and **f** pH3 (n = 2). **g-h** Initialization of E2F3A concentration with respect to **g** E2F8 (n = 3) and **h** E2F4 (n = 2). For all curves, the E2Fs intensity for the different mice are represented as curves with different line styles while EdU and pH3 are shown above the E2Fs curves, with solid lines corresponding to diffuse states and dashed lines corresponding to punctate states.
(TIF)

**S11 Fig. Initialization of E2Fs evolution over the cell cycle. a** Temporal evolution of E2F3A over the cell cycle after registration between EdU and pH3 (n = 6). **b** Temporal evolution of E2F8 over the cell cycle after registration between EdU and pH3 (n = 6). **c** Temporal evolution of E2F4 over the cell cycle after registration between EdU and pH3 (n = 4). For all curves, the E2Fs intensity for the different mice are represented as curves with different line styles while EdU and pH3 are shown above the E2Fs curves, with solid lines corresponding to diffuse states and dashed lines corresponding to punctate states.
(TIF)

**S12 Fig. Evaluation of the influence of the number of samples over the estimation of protein concentration over the cell cycle on simulated data.** Mean squared error between the estimated and simulated concentrations of E2F3A **a**, E2F8 **b**, E2F4 **c**, EdU **d** and pH3 **e** by considering different numbers of samples when corrupting up to 50% of the simulated data with noise. $n_1$ corresponds to the number of samples for E2F3A, E2F8, EdU and pH3 while $n_2$ is the number of samples for E2F4. The lines correspond to the average mean squared error while the areas represent the standard error.
(TIF)

**S13 Fig. Evaluation of the influence of the number of time bins over the estimation of protein concentration over the cell cycle on simulated data.** Mean squared error between the estimated and simulated concentrations of E2F3A **a**, E2F8 **b**, E2F4 **c**, EdU **d** and pH3 **e** by considering different numbers of time bins when corrupting up to 50% of the simulated data with noise. The lines correspond to the average mean squared error while the areas represent the standard error.
(TIF)

**S14 Fig. Evaluation of the influence of the number of intensity bins over the estimation of protein concentration over the cell cycle on simulated data.** Mean squared error between the estimated and simulated concentrations of E2F3A **a**, E2F8 **b**, E2F4 **c**, EdU **d** and pH3 **e** by considering different numbers of intensity bins when corrupting up to 50% of the simulated data with noise. The lines correspond to the average mean squared error while the areas represent the standard error.
(TIF)

**S1 Table. Time computation for training nuclei segmentation models.** Computation time needed to train the five different deep learning approaches on the four training datasets with a GeForce RTX 2080 with Max-Q design. CF stands for confocal images, WF half stands for half the widefield images, WF stands for widefield images and CFWF stands for confocal and widefield images.
(TIF)

**S2 Table. Training strategies for nuclei segmentation.** Pre-processing, normalization, optimization method, data augmentation, transfer learning and post-processing used to train the five deep learning approaches.
(TIF)

**S3 Table. Training data summary.** Number of images and nuclei in the four different training and validation datasets. CF stands for confocal images, WF half stands for half the widefield images, WF stands for widefield images and CFWF stands for confocal and widefield images.
(TIF)

**S4 Table. Time computation for processing nuclei segmentation.** Computation time needed to process the five different deep learning approaches on the four training datasets with a GeForce RTX 2080 with Max-Q design.
(TIF)

**S5 Table. Evaluation of deep learning approaches for nuclei segmentation in confocal images.** F1 score obtained for IoU = 0.5 and IoU = 0.75 when segmenting nuclei in confocal images with U-Net, Inception-V3, Mask R-CNN, Stardist and Cellpose. DA stands for data augmentation, PP stands for post-processing, TL stands for transfer learning and IoU stands for intersection over union.
(TIF)

**S6 Table. Evaluation of deep learning approaches for nuclei segmentation in widefield images.** F1 score obtained for IoU = 0.5 and IoU = 0.75 when segmenting nuclei in widefield images with U-Net, Inception-V3, Mask R-CNN, Stardist and Cellpose. DA stands for data augmentation, PP stands for post-processing, TL stands for transfer learning and IoU stands for intersection over union.
(TIF)

**S7 Table. Performance of Inception-V3 for marker identification with confocal images.** Inception-V3 accuracy and standard error with and without transfer learning for marker identification of E2F3A, E2F8, EdU and pH3 in confocal images. DA stands for data augmentation.
(TIF)

**S8 Table. Performance of Inception-V3 for marker identification with widefield images.** Inception-V3 accuracy and standard error with and without transfer learning for marker identification of E2F3A, E2F8, E2F4, EdU and pH3 in widefield images. DA stands for data augmentation.
(TIF)

**S9 Table. Time computation for training and processing marker identification.** Computation time needed to train and process Inception-V3 for marker identification with a GeForce RTX 2080 with Max-Q design. DA stands for data augmentation.
(TIF)

## Author Contributions

**Conceptualization:** Thierry Pécot, Maria C. Cuitiño.

**Data curation:** Thierry Pécot, Maria C. Cuitiño.

**Formal analysis:** Thierry Pécot, Maria C. Cuitiño, Cynthia Timmers, Gustavo Leone.

**Funding acquisition:** Gustavo Leone.

**Investigation:** Thierry Pécot.

**Methodology:** Thierry Pécot.

**Software:** Thierry Pécot.

**Supervision:** Thierry Pécot.

**Validation:** Thierry Pécot.

**Writing – original draft:** Thierry Pécot.

**Writing – review & editing:** Maria C. Cuitiño, Roger H. Johnson, Cynthia Timmers, Gustavo Leone.

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
