## [Decision Letter · Decision Letter 0]

4 Jun 2021

Dear Dr Pecot,

Thank you very much for submitting your manuscript "Deep learning tools and modeling to estimate the temporal expression of cell cycle proteins from 2D still images" for consideration at PLOS Computational Biology.

As with all papers reviewed by the journal, your manuscript was reviewed by members of the editorial board and by several independent reviewers. In light of the reviews (below this email), we would like to invite the resubmission of a significantly-revised version that takes into account the reviewers' comments.

We cannot make any decision about publication until we have seen the revised manuscript and your response to the reviewers' comments. Your revised manuscript is also likely to be sent to reviewers for further evaluation.

Sincerely,

Attila Csikász-Nagy

Associate Editor

PLOS Computational Biology

Jason Haugh

Deputy Editor

PLOS Computational Biology

Reviewer's Responses to Questions

**Comments to the Authors:**

Reviewer #1: The authors have three main contributions.

1- Benchmarking three existing computer vision architectures for cell segmentation (U-Net, Mask-R-CNN, Inception V3)

2- Benchmarking a computer vision architectures for estimating protein concentration (Inception V3)

3- Software tools that implement their pipeline (github repo + imagej plugin).

4- An optimization based approach to estimate protein concentration over time.

In part 1 and 2, the results are presented in an extremely confusing way. For example:

- At the very least, the authors should have a table, showing most of the relevant information in a single place. This would include: number of cells (not just images!) used for training, training time, prediction, etc. Figure 2 includes 2 subplots to show 16 numbers.

- It would be nice to see training curves for the different methods. The authors speculate that certain methods benefit more from data, but it would be nice to see this evaluated.

-Computer vision is a very quickly moving field, and I don't expect people in other fields to use and benchmark all the latest methods - but - it would be nice to see more methods get benchmarked. Ross Wightman has all the major methods implemented here with a consistent API and pretrained weights: https://github.com/rwightman/pytorch-image-models

- The methods are not described in nearly enough detail, and the text is difficult to follow. We should have clear diagrams that show the training pipeline for each method, which method used pre-training, which didn't, what method used post processing, etc - some of this information is scattered throughout the text, but it's a huge pain to have to go back and forth.

- The Python code (as presented in the repo) is not really packaged in a way that someone else could use it for their own use case. I appreciate the authors making the code available - but - in its current form, it's not really a resource for anyone else. At the very least, there should be a better Readme. Ideally, the code would be packaged in a library with a unified API.

- The authors discuss quantizing E2F signal in 4 buckets, but never show any kind of data justifying this decision - at the very least a histogram of concentration should be shown.

- The last part of the paper showing how the graph assignment is solved is clever, and I appreciated the author explaining why they felt justified in making certain assumptions. I wish they slightly more precise - for example - what exactly does "2 Temporal evolution of protein accumulation is similar in all observed cells." mean?

Overall, I feel the technical issues in the paper + the lack of clarity means the paper should not be accepted in its current form.

Reviewer #2: The manuscript focuses on the temporal expression of E2F proteins during the cell cycle.

In a first part, the segmentation of nuclei is performed based on a deep-learning

approach. The performance of several networks are compared, and the Mask R-CNN is chosen as

more appropriate. In a second part, the fluorescence pattern of proteins is investigated,

using a combination of deep-learning approach and manual correction of the results.

A software was developed to help correct and validate the results obtained by automated

segmentation. Finally, the temporal evolution of protein concentration is investigated by

considering a large population of cells at various stages in a static image, and reconstructing

the time variations of fluorescence between stages using graph optimization methods.

The manuscript present a number of results together with an exhaustive explanation on the

methods. However, I found it difficult to read, making it complicated to clearly identify what

were the questions addressed within the text. For instance, it is not clear if the paper aims at

solving a biological question by using computational methods, or if it focuses on the

methodological part, in particular deep-learning segmentation, using the biological question as

an illustration. This confusion deserves the presentation of the results, that however seem to

present interest.

If the biology is the main concern, I suggest putting more emphasis on it in the introduction,

and introduce the methodological questions based on it. One possibility could be to start from

the limitations encountered in ref [11], and use them to introduce the proposed enhancements.

Another option could be to focus on the methodological questions. Actually the results put

emphasis on the performance and accuracy of the methods. However this option would imply spliting

the manuscript into two distinct papers.

Other comments

* the introduction reviews segmentation of fluorescent structures approaches by considering

deep-learning approaches. While very popular, DL-approach is not the only approach for

segmentation, and discussing with respect to other "classical" approaches could help relate to

the field.

* L50-52: well, I guess everyone would expect a deep-learning approach to perform better than

a threshold approach...

* L56: it would be more clear to explain the three assumptions directly in the introduction.

* L92 (related to first point) authors use watershed to improve their segmentation workflow.

WS could be stated in the introduction.

* L116: It is not very clear to me why the authors use two different modalities. Is this to

increase the genericiy of the segmentation approach? the strategy should be better explained.

* L118 better justify relevancy of transfer learning (and of using both confocal and wide-field).

If the structures to segment are not the same, it is rather surprising to apply the same network.

Also, it is not natural to discover the use of transfer learning at his point, this could have

been stated earlier.

* L159: sentence is not clear.

* L160: this section would benefit from a better explanation of what do the author want to

obtain from the image, including an explanation / a recall about the different patterns (punctate

or diffuse) that can be observed during cycle, and eventually the associated segmentation difficulties.

* L160: I understand that thresholding may be limited for segmentation of diffuse patterns.

But many image segmentation methods exist and could have been explored. It would be interesting to

better explain the difficulty encountered, and better justify why DL seems to be the most adequate.

* L177: it is not clear if the "Annotater" plugin is a side-product of the research, or if the

whole workflow rely on it. This could be clarified.

* L204: maybe recall why it is interesting to focus on protein concentration. the transition from

previous paragraph is abrupt.

* L 236: concentrations are not parabola... The representation of their evolution may depict a

parabola, however.

* L360-367: the note at the nd of the paragraph would be more appropriate in the result or in the

discussion section.

* L492-615: this section if quite technical, and it is rather surprising to have such an inbalance

between the method description and the exploitation of the results. I wonder if this modelling part

can not be an article of itself?

Reviewer #3: This manuscript by T. Pecot and al. propose a workflow, based on deep learning (DL) segmentation and modeling, to reconstruct time courses of protein accumulation from fixed images. This work is timely and important for two reasons. First, while DL has been shown to work well for biological images, most reported uses consider proof of principle, standard datasets, or simple cases; reporting usage in full fledged, biologically relevant, and non trivial example is important and useful. Second in vivo video microscopy is costly and difficult, when it is possible, and recovering time courses from fixed samples in a reliable and efficient way is also important and useful. Overall the paper is well written, the study is well conducted, and the methods used, evaluations done and results reported are convincing.

However it seem to me to lack a clear purpose, by attempting to be too many things at once and not quite succeeding properly at any of them. Is it an evaluation of standard deep learning architecture for nucleus segmentation in non trivial images? a report on active/interactive learning - human in the loop workflow to try and address the labeling issue for biological images, by introducing a new imagej plugin? or really the presentation of a new workflow of time course estimation from from fixed image ? According to the title (and indeed, in my humble opinion as well, since it is the most novel and innovative) it should be the later; however it is the least investigated and evaluated part, as opposed to DL, with one unique graph in the main paper and very little discussion. Suggestions could be:

- to study the influence of DL segmentation accuracy for time course estimation, providing evaluation of the workflow as a whole

- additional evaluation of the estimated time courses. Depending on the availability/feasibility of experiments, those could be done with simulation. At the very least corroboration of the found time courses from other experimental or biological arguments.

- application of the same technique on other published data

- provide some novel understanding or biological findings those data/methods allowed

- discuss some of the techniques that has been proposed for similar ends (https://doi.org/10.1038/nmeth.3545 or [https://doi.org/10.1038/s41467-017-00623-3](https://doi.org/10.1038/s41467-017-00623-3) comes to mind for example).

Other points:

- the DL methods are not compared with current/recent SOTA, like cellpose or stardist. Reporting their use (and potentially failure) would be interesting to the community

- The efficacy of deep learning for segmenting and classifying biological images is now well accepted and it may not be needed to insist on it to much

- The article talks about 'a small training dataset' but still many tens of images had to be manually segmented. Maybe a study of the influence of training dataset size on accuracy could be interesting, and/or a comparison of

- The assumption that "fluorescence intensity is proportional to protein concentration" could be discussed further, since it is not so obvious in IF. [30] seem to be about a novel technique for in vivo expression, it is not clear how it relates to that claim.

- Maybe a personal preference but bar plot + whiskers may not be the ideal way to report such results. For mean+-standard deviation, simple tables may suffice. Graphical representation could be more efficient when used to display whole distributions, like violin plots or similar techniques.

- more generally, unless I missed it, it seem the paper has been submitted as a research article and not specifically as a method paper as it is, it may lack novel biological findings for that, but would certainly have its place as a method paper, the comments above notwithstanding.

**Have the authors made all data and (if applicable) computational code underlying the findings in their manuscript fully available?**

Reviewer #1: Yes

Reviewer #2: Yes

Reviewer #3: Yes

PLOS authors have the option to publish the peer review history of their article (what does this mean?). If published, this will include your full peer review and any attached files.

Reviewer #1: No

Reviewer #2: No

Reviewer #3: No
---

## [Decision Letter · Decision Letter 1]

8 Jan 2022

Dear Dr Pécot,

Thank you very much for submitting your manuscript "Deep learning tools and modeling to estimate the temporal expression of cell cycle proteins from 2D still images" for consideration at PLOS Computational Biology.

As with all papers reviewed by the journal, your manuscript was reviewed by members of the editorial board and by several independent reviewers. In light of the reviews (below this email), we would like to invite the resubmission of a significantly-revised version that takes into account the reviewers' comments.

The errors in the upload of Fig 2 and Fig 3 made it hard to follow the exact changes. Some of these have improved the manuscript. But several major points are still remains unanswered, thus another round of majr revision on the MS is needed. 

We cannot make any decision about publication until we have seen the revised manuscript and your response to the reviewers' comments. Your revised manuscript is also likely to be sent to reviewers for further evaluation.

Sincerely,

Attila Csikász-Nagy

Associate Editor

PLOS Computational Biology

Jason Haugh

Deputy Editor

PLOS Computational Biology

Reviewer's Responses to Questions

**Comments to the Authors:**

Reviewer #1: - I thank the authors for taking the time to review my feedback. I found tables S2 and S4 quite helpful to understand the different training pipelines. I would like to see a similar table when comparing the performance of the different methods: I would suggest a given IoU score (or, multiple) - and then comparing performance of all the different methods and showing those numbers in a table. Having to jump between tens of different subfigures to see the impact of data augmentation is very difficult and time consuming. I understand that a table cannot summarize all the results across the different IoU thresholds, and the authors can keep the figures in the supplementary material if necessary, but, it's really necessary to have a single table with the important performance metrics.

- Figure 2 and 3 are identical in my copy of the manuscript, and I could not evaluate that portion of the manuscript because I couldn't see the data.

- I still insist that if the authors would like to bin E2F data, they should show the raw histogram data justifying a binning procedure. They have that raw data (they need it to obtain the binned values) - and plotting a histogram, together with the cutoff thresholds should be very easy. I understand that this is standard in their field - but - for someone outside of the field looking to evaluate the manuscript, continuous data getting binned without seeing the raw distributions is confusing.

- The suggestions from reviewer 2 about presentation and multiple papers are great. Currently- this is two papers, and one paper might be better suited for a computer vision conference or workshop, while the biological findings and applications might belong here. For example - training curves and detailed comparisons to other methods would be really useful if this was a pure computer vision paper, but, in a biology paper, they are probably overkill, as people are not that interested in evaluating the methods at that level of detail.

Reviewer #2: The authors have taken into account most of the remarks made in previous version.

The manuscript is still very complicated to read, and it still lacks clarity of the objectives. However, I have no additional remark to bring and I consider it can be published as is.

Reviewer #3: The resubmitted article is much improved compared to the first version and in particular more focused. I have no major request of changes to be made before publication, just a few comments below.

The figures are wrong in the submitted manuscript (fig 2 and 3 are the same and Fig 4 seem unrelated). A bit problematic for the review but the main novelty are the simulations, which are in the supplementary. To be thoroughly checked before publication though...

Fig 3 has only the final results with all simulation and method detail relegated in the supplementary. I personally think it's a bit of a shame since the figures are the main points of entry for many reader, which would make that work invisible. I would welcome some of that in the main figures, but leave that choice to the authors/editors.

The simulations have pretty big error bars, and the noise seem to be only marginally affecting the results in many cases with averages being fairly constant from 0 to 30 and often 50% noise. Can the authors comment on that? Make one wonder if that is the most informative way of displaying that result, and what would the error bar on real data would be. Also the simulations are presented in the text before the time course estimation methods, while it uses/is meant to test it, which is a bit strange.

**Have the authors made all data and (if applicable) computational code underlying the findings in their manuscript fully available?**

Reviewer #1: Yes

Reviewer #2: Yes

Reviewer #3: Yes

PLOS authors have the option to publish the peer review history of their article (what does this mean?). If published, this will include your full peer review and any attached files.

Reviewer #1: No

Reviewer #2: No

Reviewer #3: No
---

## [Decision Letter · Decision Letter 2]

14 Feb 2022

Dear Dr Pécot,

Thank you very much for submitting your manuscript "Deep learning tools and modeling to estimate the temporal expression of cell cycle proteins from 2D still images" for consideration at PLOS Computational Biology. As with all papers reviewed by the journal, your manuscript was reviewed by members of the editorial board and by several independent reviewers. The reviewers appreciated the attention to an important topic. Based on the reviews, we are likely to accept this manuscript for publication, providing that you modify the manuscript according to the review recommendations.

Please explain the method of thresholding, as requested by Ref. 1.

Sincerely,

Attila Csikász-Nagy

Associate Editor

PLOS Computational Biology

Jason Haugh

Deputy Editor

PLOS Computational Biology

[LINK]

Reviewer's Responses to Questions

**Comments to the Authors:**

Reviewer #1: I thank the authors for all the additions since the last review. I found the extra tables especially helpful, and appreciated the additional figure showing the E2F concentration.

In my view, the E2F quantization step is not justified on the basis of data: the distribution in figure S7 is obviously not multi-modal, and, in the material and methods section, the thresholding step was described as using 'manually defined thresholds', and there's no discussion at all about how those thresholds were calculated or how sensitive the results later in the paper are to the choice of this arbitrary threshold.

It might be that there is a biological justification for this thresholding step - and - at the very least, I'd like the authors discuss this in more detail in the main text. Currently, the main text has a quote:

"We assume that fluorescence intensity is proportional to protein =concentration [34, 35] and therefore define quantized levels of intensity for E2Fs."

Clearly, the second part (deciding to quantize a signal) does not follow from the first (protein concentration being proportional to the fluorescent signal).

Reviewer #3: All my comment on the text and figures have been addressed.

**Have the authors made all data and (if applicable) computational code underlying the findings in their manuscript fully available?**

Reviewer #1: Yes

Reviewer #3: Yes

PLOS authors have the option to publish the peer review history of their article (what does this mean?). If published, this will include your full peer review and any attached files.

Reviewer #1: No

Reviewer #3: No

Figure Files:

Data Requirements:

Reproducibility:

References:

---

## [Editor Report · Decision Letter 3]

21 Feb 2022

Dear Dr Pécot,

We are pleased to inform you that your manuscript 'Deep learning tools and modeling to estimate the temporal expression of cell cycle proteins from 2D still images' has been provisionally accepted for publication in PLOS Computational Biology.

Best regards,

Attila Csikász-Nagy

Associate Editor

PLOS Computational Biology

Jason Haugh

Deputy Editor

PLOS Computational Biology

---

## [Editor Report · Acceptance letter]

8 Mar 2022

PCOMPBIOL-D-21-00751R3 

Deep learning tools and modeling to estimate the temporal expression of cell cycle proteins from 2D still images

Dear Dr Pécot,

I am pleased to inform you that your manuscript has been formally accepted for publication in PLOS Computational Biology. Your manuscript is now with our production department and you will be notified of the publication date in due course.

With kind regards,

Zsofia Freund
